# Perinatal mortality after Chornobyl in contaminated regions of Ukraine

**Alfred Körblein** *

Independent Researcher, Nuremberg, Germany

* alfred.koerblein@gmx.de

## Abstract

### Background

Belarus and Ukraine were the countries most affected by the consequences of the Chornobyl nuclear power plant in 1986. A study of perinatal mortality in Belarus found a highly statistically significant increase in the 1990s in the most contaminated oblast Gomel but no increase during this period in the rest of Belarus. As a possible mechanism to interpret this increase as a late Chornobyl effect, it has been suggested that strontium-90 contained in Chornobyl fallout, incorporated during menarche, impairs the immune system of pregnant women which in turn increases perinatal mortality. In the present study, this hypothesis is tested using data from Ukraine.

### Methods

Annual data on perinatal mortality, in the period 1981–2006 were provided by the Ministry of Public Health of Ukraine. Trends in perinatal mortality rates in the most contaminated regions of Ukraine (Kyiv and Zhytomyr oblasts and the city of Kyiv; study region) were compared with rates in the rest of Ukraine (control region). To identify any differences in perinatal mortality trends between the study and control regions, the ratios of perinatal mortality rates in the study region to the rates in the control region were analyzed using the calculated strontium concentration as a predictor.

### Results

A trend analysis of perinatal mortality rates in Ukraine revealed two bell-shaped deviations from a long-term exponential trend with maxima at the beginning and end of the 1990s. The same pattern was found in the data from the study and control regions, but the deviations were almost three times higher in the study region than in the control region. An analysis of the ratios of perinatal mortality rates in the study region to the rates in the control region (odds ratios) showed an increase and decrease during the 1990s which was approximated by a lognormal density distribution. The calculated strontium concentration, when used as a predictor, also fitted the data well. Thus, the data from Ukraine confirms the results from Belarus. The analysis of the odds ratios revealed about 1000 excess perinatal deaths in the study region in the period 1990–2004. The corresponding figure for Ukraine as a whole was estimated at 3500 perinatal deaths.

**Data Availability Statement:** All relevant data are within the manuscript and its Supporting Information files.

**Funding:** The author(s) received no specific funding for this work.

**Competing interests:** The author has declared that no competing interests exist.

## Conclusion

It is hypothesized that the observed increase in perinatal mortality in the 1990s may be a late effect of incorporated strontium-90 on the immune system of pregnant women. The analysis is based on a theoretical model, as no data on strontium concentrations were available; the results should therefore be interpreted with caution. An updated study for Belarus would be desirable to corroborate the results.

## Introduction

The health effects of the Chornobyl accident in 1986 were assessed in the UNSCEAR report of 2008, Annex D [1]. However, a search in the text for the term "perinatal mortality" or "infant mortality" yielded no hits. Pregnancy outcomes in Ukraine after Chornobyl were presented in the 2006 WHO report [2]. The WHO expert panel found that the studies were "largely descriptive, showing only percentage changes, with no indication of the time period or actual numbers affected." It concluded that it was "not able to evaluate the evidence and draw conclusions" (page 86 in [2]). No data on perinatal mortality, but data on infant mortality in Ukraine are presented as pictures (Figures 5–9 of Chapter 6 in [2], S1 Fig in S1 File).

In 2003, the author noted an unexpected increase in perinatal mortality in the Gomel region of Belarus between 1990 and 1998 [3]. He suggested that this increase could be a late effect of incorporated strontium on the immune system of pregnant women. His strontium model is explained in detail in [3] but will be briefly presented below. The model was also applied to analyze the trend in perinatal mortality in West Germany after the atmospheric nuclear weapon tests in the 1950s and 1960s [4].

Strontium concentration in pregnant women was approximated in each year after Chornobyl by the proportion of women who were 13 or 14 years old in 1986, the year of the Chornobyl accident. Strontium uptake is highest around the age of menarche, the time of maximum bone growth. The proportion of pregnant women aged 13 or 14 years in 1986 can be determined in each year after Chornobyl using maternal age distribution. Strontium impairs the immune system [5], which could be responsible for increased perinatal mortality for many years after the Chornobyl accident.

In June 2006, at a conference in Kyiv titled "Health consequences of the Chernobyl catastrophe. Strategy of recovery", the author presented the results of his study on perinatal mortality in Belarus after Chornobyl. The conference was organized by the Ministry of Public Health of Ukraine and the Kyiv-based association "Physicians of Chernobyl". After the conference, the author contacted Prof. Omelyanets, the head of the Laboratory of Medical Demography of Ukraine and the main author of an extensive report titled "Determination of infant mortality and morbidity in the population of Ukraine affected by the CHNPP accident". This report was compiled as part of the "Franco-German Chernobyl Initiative". Among other things, the changes in infant mortality and perinatal mortality in the contaminated Ukrainian oblasts of Zhytomyr and Kyiv and their most heavily contaminated districts were examined in comparison with the rates in the "uncontaminated" oblast of Poltava (average surface contamination less than 37 kBq/m$^2$). The author asked Professor Omelyanets for data on perinatal mortality, i.e., the annual number of live births, stillbirths, and early neonatal deaths. Collaboration was agreed on the condition that the results would be published jointly. All necessary data were sent to the author by e-mail. A joint article was prepared and submitted to the "International Journal of Radiation Medicine", the official journal of the NGO "Physicians of Chernobyl".

After review, the article was accepted for publication. Unfortunately, the journal was discontinued shortly afterward due to a lack of funding, so the article was never published.

In the present study, the data on perinatal mortality in Ukraine are analyzed according to the method described in [3]. The study aims to investigate whether the data on perinatal mortality in Ukraine provide similar results to the data from Belarus. Since the population of Ukraine is much larger than that of Belarus and the study period is six years longer (up to 2004 instead of 1998), the statistical power of the present study is much greater than that of the study for Belarus.

## Materials and methods

Annual data on perinatal mortality i.e. the numbers of live births (LB), stillbirths (SB), and early neonatal deaths (NEO), for Kyiv city, Kyiv oblast, and Zhytomyr oblast, in the period 1981–2006 were provided by the State Committee of Statistics of Ukraine and the Ministry of Public Health of Ukraine.

The perinatal mortality rates in the oblasts Zhytomyr, Kyiv, and Kyiv city (study region) are compared with the rates in the remainder of Ukraine (control region). The study- and control regions differ in radiation exposure but are assumed to be similar in their socio-economic status. Any differences in the trends can therefore be attributed to differences in radiation exposure.

The trend of perinatal mortality $y(t)$ in Ukraine is modeled by a long-term linear-quadratic exponential trend superimposed by two bell-shaped excess terms (lognormal density distributions):

$$y(t) \sim \exp(\beta_0 + \beta_1 \cdot t + \beta_2 \cdot t^2 + \beta_3/t/exp((log(t) - \beta_4)\hat{}2/2/\beta_5^2) + \beta_6/t/exp((log(t) - \beta_7)\hat{}2/2/\beta_8^2)) \tag{1}$$

Here, time t is calendar year minus 1980; parameter $\beta_0$ is the intercept, and $\beta_1$ and $\beta_2$ are the linear and quadratic trend parameters. Lognormal density distributions are used for the bell-shaped excess terms with parameters $\beta_3$ to $\beta_5$ and $\beta_6$ to $\beta_8$. The parameters $\beta_3$ and $\beta_6$ estimate the effect sizes, $\beta_4$ and $\beta_7$ estimate the logarithms of the medians, and $\beta_5$ and $\beta_8$ the standard deviations. Iteratively reweighted non-linear regression with function $nls()$ of the statistics package R (www.r-project.org) is applied with weights $1/var$ where the binomial variance $var$ is $fit \cdot (1-fit)/(LB+SB)$ and $fit$ is the fitted value.

To determine any differences in trends, the ratios of mortality rates in the study region to the rates in the control region were analyzed. Instead of the rate ratios, however, the odds ratios were used, which are defined by OR = $y_1$ / (1-$y_1$) / ($y_0$ / (1-$y_0$)). Here, $y_1$ = (SB$_1$ + NEO$_1$) / (LB$_1$ + SB$_1$) and $y_0$ = (SB$_0$ + NEO$_0$) / (LB$_0$ + SB$_0$) are the perinatal mortality rates in the study region (subscript 1) and the control region (subscript 0). For small values of $y_1$ and $y_0$, the differences between rate ratios and odds ratios are negligible. The natural logarithms (ln) of the odds ratios are used here because they allow simple calculation of the variances required for the weighted regression. The regression model has the following form:

$$ln(OR) \sim \beta_0 + \beta_1 \cdot t + \beta_2/t/exp((log(t) - \beta_3)\hat{}2/2/\beta_4^2) \tag{2}$$

Parameter $\beta_0$ estimates any difference in the level of perinatal mortality between the study and control regions, and $\beta_1$ allows for a time trend in the odds ratios. The increase in odds ratios after Chornobyl is modeled by a bell-shaped excess term (parameters $\beta_2$ through $\beta_4$) which is suggested by the data. Weighted non-linear regression is used with weights $1/var$, where

variance has the following form:

$$var = 1/(SB_1 + NEO_1) + 1/(LB_1 - NEO_1) + 1/(SB_0 + NEO_0) + 1/(LB_0 - NEO_0)$$

The number of excess cases in a given year k is calculated as $O[k]/OR[k] \cdot exp(\beta_1 + \beta_2 \cdot t[k])$, where $O[k]$ is the observed number of infant deaths, $OR[k]$ is the observed odds ratio and $exp(\beta_1 + \beta_2 \cdot t[k])$ is the predicted undisturbed odds ratio in year k. The 95% confidence interval for the number of excess cases is determined from the 95% confidence limits of the parameter $\beta_2$ in the model (1) and from the 95% confidence limits of the parameter $\beta_3$ in model (2). A two-sided p-value $p < 0.05$ is considered statistically significant.

## Strontium model

The strontium in the fallout from Chornobyl replaces calcium in the bones. The maximum uptake of strontium into the bones occurs during the period of major bone growth which, for girls, is around the age of 13 to 14 years, see S2 Fig in S1 File. To simplify the calculation, it is assumed that the average strontium concentration in the cohort of pregnant women in a given year after Chornobyl is determined only by the proportion of women who were 13 or 14 years old in 1986. This proportion follows from the age distribution of the mothers (maternal age distribution). Unfortunately, the author did not have any Ukrainian data on the age distribution of mothers. However, at a conference in Saint Petersburg in 2001, the author obtained annual data on maternal age distribution from Saint Petersburg which were used as a proxy in the present study.

The strontium concentration $Sr(x)$ in year x after 1986 depends on the proportion P(age) of mothers of age x+13 or x+14, adjusted by a strontium elimination rate of 2.6% per year for women aged 20–40 years [6]. For an age of 14 years at exposure, the strontium term has the following form:

$$Sr(x) = P(x + 14) \cdot exp(-0.026 \cdot x)$$

To analyze the odds ratios with the strontium term, the lognormal function in the model (2) is replaced by $Sr(x)$:

$$ln(OR) \sim \beta_1 + \beta_2 \cdot t + \beta_3 \cdot Sr(x). \tag{3}$$

## Results

Table 1 shows the summary statistics of live births (LB), stillbirths (SB), and early neonatal deaths (NEO) for the periods 1981–1989 and 1990–2004 in Ukraine and the study and control regions, where the study region includes Kyiv and Zhytomyr oblasts and the city of Kyiv. The control region is Ukraine excluding the study region.

Fig 1 shows the development of perinatal mortality rates in Ukraine in the period 1981–2006 together with the perinatal mortality rates in Poland and Germany. Until the mid-1990s,

**Table 1. Summary statistics of perinatal morality in Ukraine.**

| region | 1981–1989 | | | 1990–2006 | | |
|---|---|---|---|---|---|---|
| | LB | SB | NEO | LB | SB | NEO |
| Ukraine | 6,829,157 | 66,285 | 38,065 | 8,049,505 | 52,774 | 41,752 |
| study region | 799,567 | 7,800 | 3,738 | 942,792 | 6,303 | 5,173 |
| control region | 6,029,590 | 58,485 | 34,327 | 7,106,713 | 46,471 | 36,579 |

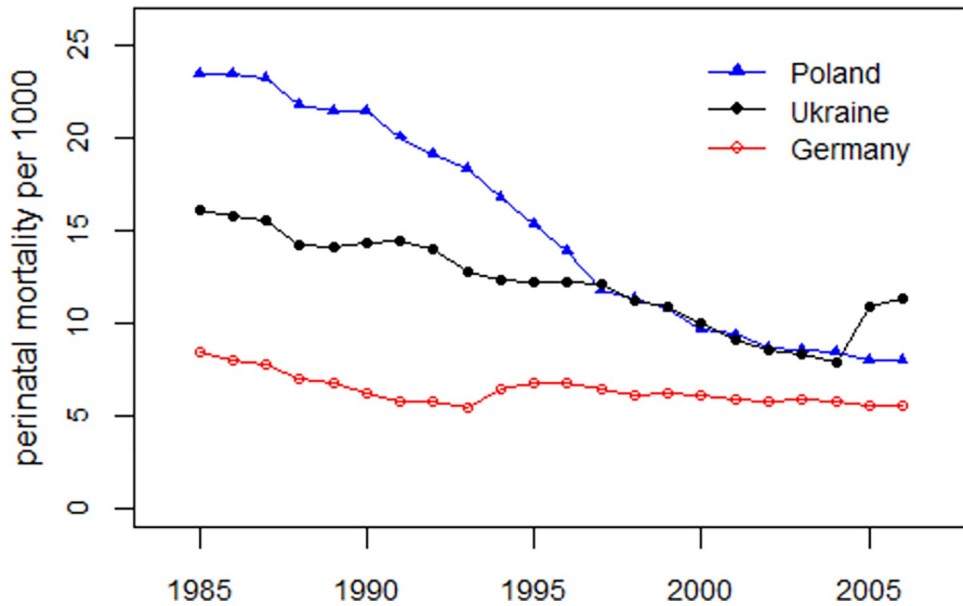

**Fig 1. Development of perinatal mortality rates in Poland, Ukraine, and Germany, 1981–2006, with level shifts in 2005 in Ukraine and 1994 in Germany.**

there were large differences in the levels of perinatal mortality with the highest level in Poland and the lowest in Germany.

The development over time of perinatal deaths and the total number of deaths, i.e., live births plus stillbirths in Ukraine, is shown in S3 Fig of S1 File. The number of perinatal deaths increased slightly in 1981–1986 and declined after 1986. The number of total births shows a similar pattern, with a slight increase until 1986, a steady decline thereafter, and a slight increase in 2000–2004. S4 Fig in S1 File shows the development of the corresponding data from the oblasts Kyiv, Zhytomyr, and the city of Kyiv.

## Trend analyses

**Ukraine.** The search for an appropriate regression model to fit the perinatal mortality rates in Ukraine started with an exponential trend with a linear-quadratic time dependence suggested by the data. The data analysis was limited to the years 1981–2004, as there was a highly significant increase in the years 2005 and 2006 compared to the trend in the pre-2005 data, maybe due to a change in the definition of stillbirth. In 1987, a significant 5% peak in perinatal mortality was observed in Germany [7,8]; a dummy variable was therefore added to the regression model to account for a possible increase in 1987. The regression model fitted the data reasonably well albeit with large overdispersion; the deviance was 182.6 at 20 degrees of freedom (df = 20). Fig 2, panel A, shows the trend in perinatal mortality rates in Ukraine and the regression result. Panel B shows the standardized residuals, i.e., the deviations of the perinatal mortality rates from the fitted model in units of standard deviations. The residuals oscillate around the zero line. There is no discernible influence of the Chornobyl accident on perinatal mortality.

At the above-mentioned conference in Saint Petersburg, the author was warned not to use data before 1985 (Natalia Kovaleva, private communication). Before Gorbachev's glasnost policy, health data from the Soviet Union were probably underreported. Regression without the pre-1985 data reduced the deviance from 183 (df = 20) to 141 (df = 16). However, due to large

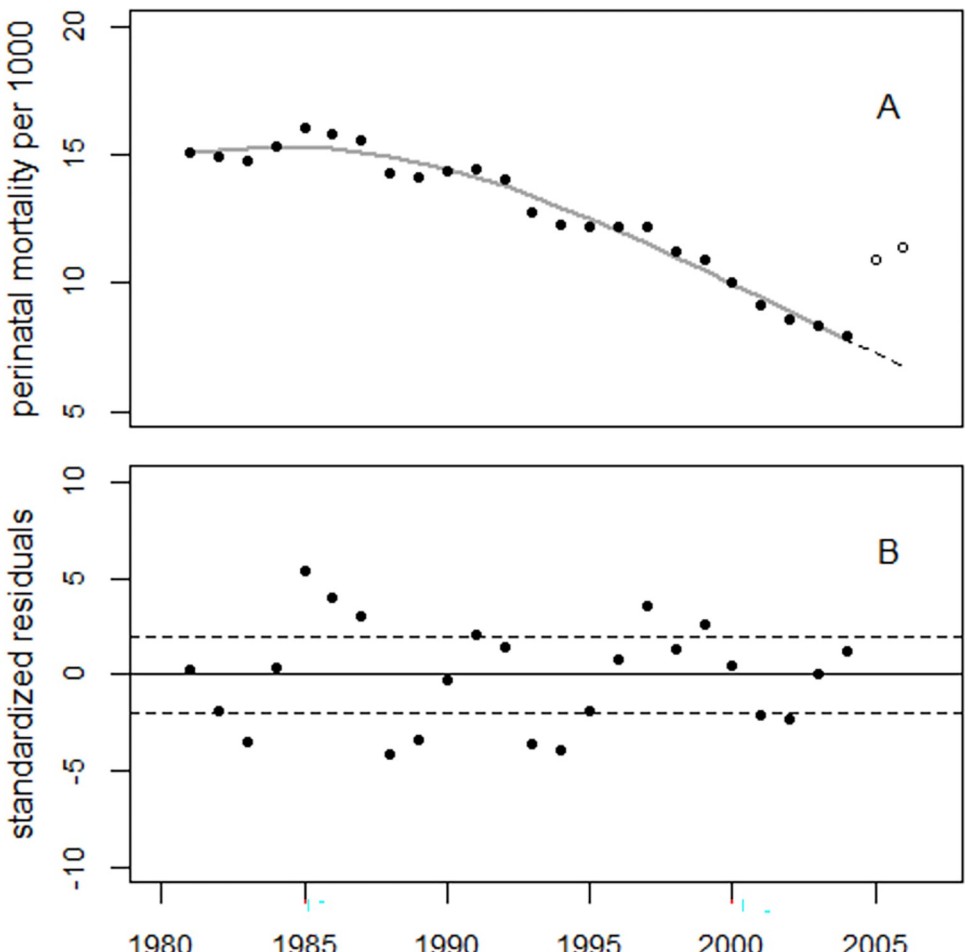

**Fig 2.** Panel A: Perinatal mortality rates in Ukraine and regression line. Panel B shows the deviations of the observed rates from the fitted trend in units of standard deviations (standardized residuals).

overdispersion, the improvement in fit was not statistically significant (p = 0.36, F-test). Since the trend of the residuals peaked at the beginning and end of the 1990s, two bell-shaped excess terms (lognormal density distributions) were added to the regression model. The two excess terms yielded a highly significant improvement in the fit; the deviance decreased from 141 (df = 20) to 13.2 (df = 10). Next, a common half-width was used since there was no notable difference between the half-widths of the two lognormal distributions (p = 0.76). Finally, the squared time variable (t2) was removed from the regression model as it did not differ appreciably from zero (p = 0.47). The dispersion factor decreased from OD = 9.13 with the first model to a moderate value of OD = 1.16 with the final model. The improvement in model fit with each step of model refinement is shown in Table 2.

Fig 3 shows the development of perinatal mortality rates in Ukraine in the period 1981–2006 and the result of the regression with the modified model (1). Before 1985, the perinatal mortality rates were significantly lower than the rates predicted from the trend after 1984 (see dashed line in Fig 3).

The results of regression with the final model, i.e., model (1) without the quadratic time variable and with equal widths of the bell-shaped excess terms, are shown in Table 3. The parameter $\beta_8$ estimates the increase in 1987, the year after the Chornobyl accident. For the period 1990–

**Table 2. Improvement in model fit from stepwise addition of variables.**

| variables | deviance | df2 | OD[1] | df1 | F-value | p-value |
|---|---|---|---|---|---|---|
| time t1, t2 | 182.65 | 20 | 9.13 | | | |
| minus 1981–84 | 140.98 | 16 | 8.81 | -4 | 1.18 | 0.356 |
| excess terms | 13.19 | 10 | 1.32 | -6 | 16.15 | <0.001 |
| common half-width | 13.31 | 11 | 1.21 | 1 | 0.10 | 0.761 |
| removing t2 | 13.98 | 12 | 1.16 | 1 | 0.55 | 0.474 |

[1]OD (for overdispersion) is deviance divided by degrees of freedom (df2).

2004, the number of excess perinatal deaths was estimated at 7,284 (95% CI: 6,373 to 8,520) which corresponds to an average increase of 9.4 (8.1 to 11.2) percent. In 1987, there was a statistically significant increase of 3.8 (1.3 to 6.3) percent corresponding to 435 (145 to 727) excess perinatal deaths. The annual numbers of live births (LB), stillbirths (SB) and early neonatal deaths (NEO), perinatal mortality rates, and fitted values are shown in S3 Table of S2 File.

In addition to perinatal mortality, data on the stillbirth rate and early neonatal mortality from 1985 to 2004 were also analyzed. The parameter estimates with standard errors (SE) and

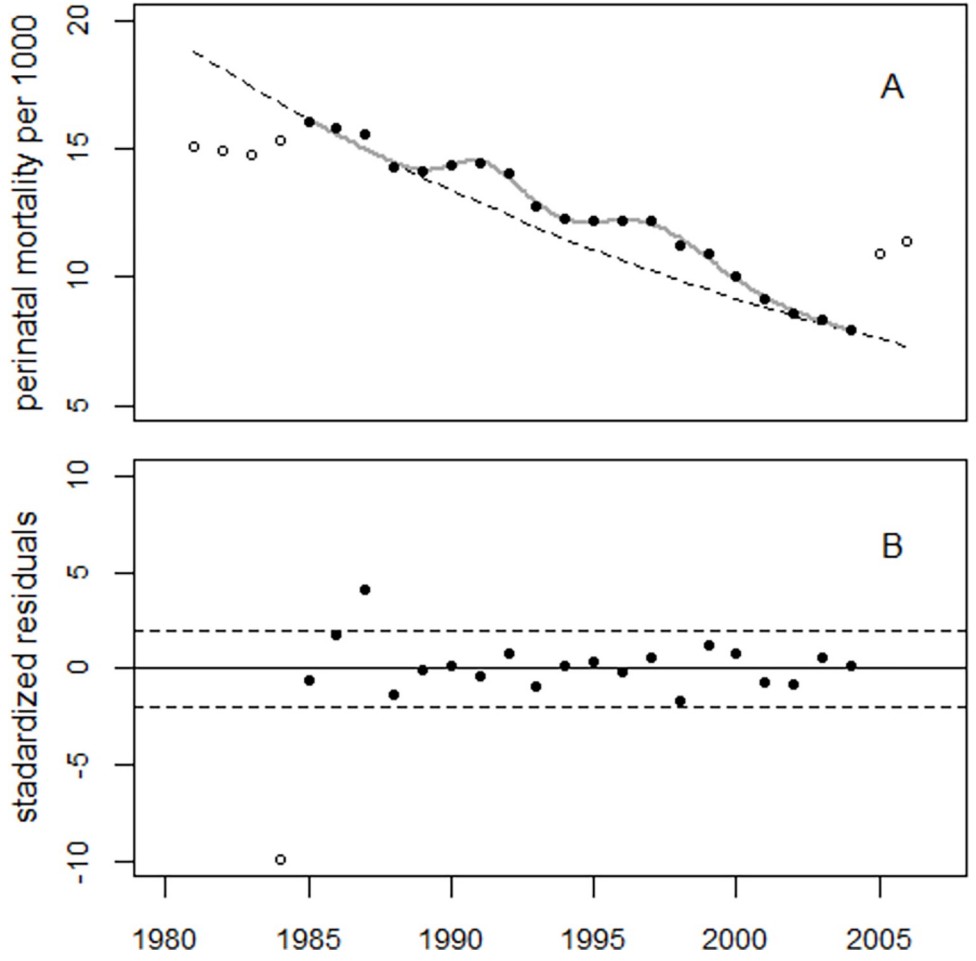

**Fig 3.** Panel A: Perinatal mortality rates in Ukraine, 1981–2006, result of regression of 1985–2004 data (black circles). Panel B: Deviations of observed rates from the fitted trend in units of standard deviations (standardized residuals).

**Table 3. Regression results for perinatal mortality rates from Ukraine.**

| Parameter | Estimate | SE[1] | t-value | p-value |
|---|---|---|---|---|
| $\beta_0$ | -3.936 | 0.009 | -454.2 | <0,0001 |
| $\beta_1$ | -0.0377 | 0.0008 | -44.96 | <0,0001 |
| $\beta_3$ | 1.413 | 0.112 | 12.57 | <0,0001 |
| $\beta_4$ | 2.445 | 0.011 | 218.3 | <0,0001 |
| $\beta_5$ | 0.121 | 0.008 | 14.68 | <0,0001 |
| $\beta_6$ | 2.816 | 0.228 | 12.33 | <0,0001 |
| $\beta_7$ | 2.864 | 0.011 | 251.5 | <0,0001 |
| $\beta_8$ | 0.037 | 0.011 | 3.30 | 0.0063 |

[1]SE: Standard error.

t-values are shown in Table 4. The model fitted both data well, see Fig 4. The deviance was 24.8 (df = 12) for stillbirths and 18.2 (df = 12) for early neonatal deaths. The number of excesses in the period 1990–2004 was estimated at 1,947 (1,402 to 2,705) stillbirths, an overall increase of 6%, and 6,539 (5,295 to 8,884) early neonatal deaths, an overall increase of 32%. The peak in 1987 was statistically significant for stillbirths (+4.3%; p = 0.047), but not significant for early neonatal deaths (+3.6%; p = 0.127).

**Study and control regions.** Data on early neonatal mortality from Kyiv city were only available from 1985 onwards so the trend of perinatal mortality in the entire study region was analyzed for the period 1985–2004. The results of the regressions of the trends of perinatal mortality in the study and control regions with two bell-shaped excess terms are shown in Table 5 and Fig 5. The increase in perinatal mortality after 1989 was almost three times as high in the study region as in the control region. For the period 1990–2004, 1,792 (1,512 to 2,406) excess perinatal deaths were estimated in the study region. In 1997, there was a conspicuous peak in the data from the study region. Omitting this data point reduced the deviance substantially from 13.8 (df = 12) to 6.9 (df = 11), p = 0.007 (F-test). The data point was therefore considered an outlier and removed from the regression. As a result, the number of excess deaths in the period 1990–2004 increased slightly to 1,882 with a lower limit of 1,589 (the calculation of the upper limit with the R function *confint*() failed; the number of iterations exceeded the maximum of 50). The corresponding result for the control region was 5,484 (4,566 to 6,840) excess cases, an average increase of 8.0% (6.6% to 10.2%) over the period 1990–2004. In 1987, perinatal mortality was significantly increased in both regions, with 80 (10 to 152) excess cases (+6.5%, p = 0.028) in the study region and 363 (55 to 674) excess cases (+3.5%, p = 0.024) in the control region (see last row of Table 5). The annual numbers of live births, stillbirths, and

**Table 4. Regression results for early neonatal mortality and stillbirth rate in Ukraine.**

| Parameter | Early neonatal mortality | | | Stillbirth rate | | |
|---|---|---|---|---|---|---|
| | Estimate | SE | t-value | Estimate | SE | t-value |
| $\beta_0$ | -4.950 | 0.020 | -247.8 | -4.364 | 0.014 | -308.8 |
| $\beta_1$ | -0.033 | 0.003 | -12.74 | -0.044 | 0.001 | -36.40 |
| $\beta_3$ | 2.724 | 0.234 | 11.66 | 0.801 | 0.191 | 4.20 |
| $\beta_4$ | 2.487 | 0.013 | 187.0 | 2.391 | 0.028 | 86.51 |
| $\beta_5$ | 0.145 | 0.012 | 12.11 | 0.097 | 0.021 | 4.53 |
| $\beta_6$ | 5.367 | 0.640 | 8.38 | 1.602 | 0.360 | 4.44 |
| $\beta_7$ | 2.909 | 0.018 | 157.4 | 2.839 | 0.025 | 112.5 |
| $\beta_8$ | 0.035 | 0.021 | 1.64 | 0.042 | 0.019 | 2.21 |

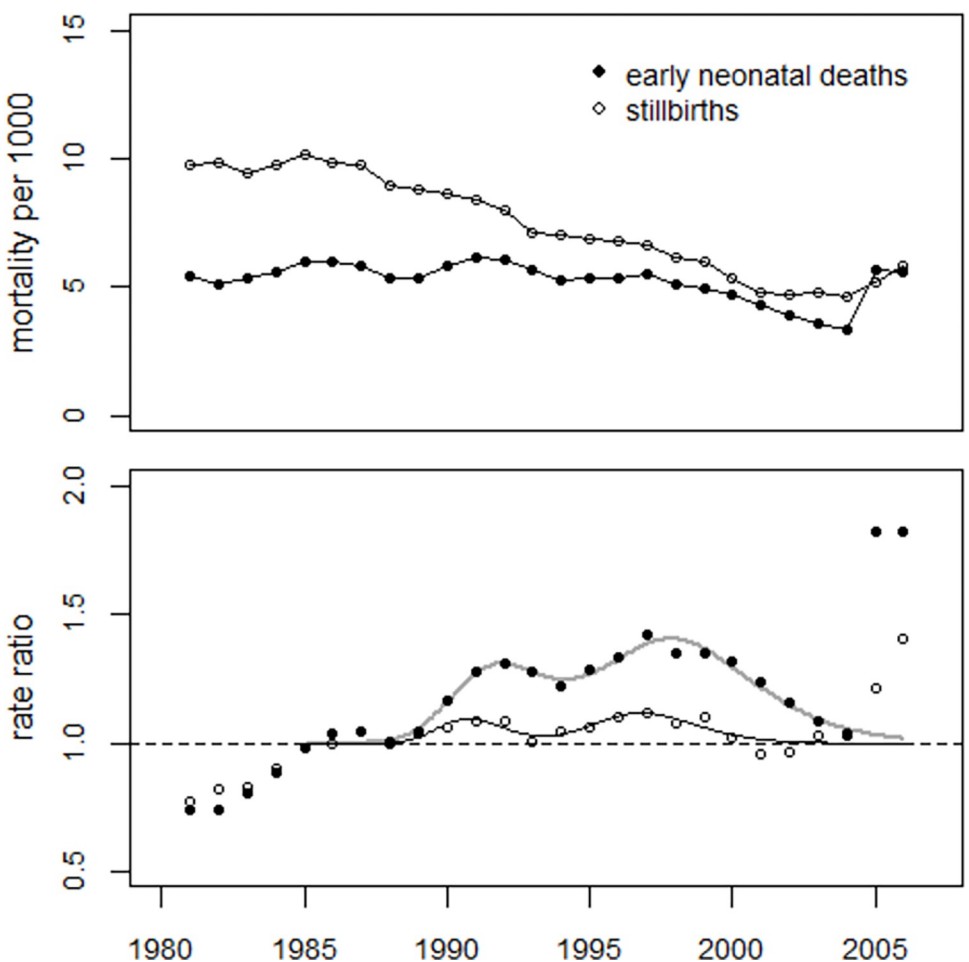

**Fig 4. Upper panel: Development of early neonatal mortality and stillbirth rates in Ukraine Lower panel: Ratio of observed to undisturbed predicted rates (rate ratios).**

early neonatal deaths in the study region and the control region, together with the perinatal mortality rates and the fitted values, are shown in S2 and S3 Tables of S2 File.

**Analyses at the oblast level.** The annual numbers of live births (LB), stillbirths (SB) and early neonatal deaths (NEO) in Kyiv City and the oblasts Kyiv and Zhytomyr are shown in S1

**Table 5. Regression results for perinatal mortality in the study- and control regions.**

| Parameter | Study region | | | Control region | | |
|---|---|---|---|---|---|---|
| | Estimate | SE | t-value | Estimate | SE | t-value |
| $\beta_0$ | -3.861 | 0.021 | -185.6 | -3.946 | 0.010 | -381.2 |
| $\beta_1$ | -0.0455 | 0.0025 | -18.03 | -0.0368 | 0.0010 | -36.94 |
| $\beta_3$ | 3.013 | 0.320 | 9.40 | 1.243 | 0.133 | 9.34 |
| $\beta_4$ | 2.520 | 0.021 | 122.7 | 2.431 | 0.014 | 168.5 |
| $\beta_5$ | 0.148 | 0.018 | 8.13 | 0.117 | 0.011 | 10.66 |
| $\beta_6$ | 4.226 | 0.583 | 7.25 | 2.527 | 0.278 | 9.10 |
| $\beta_7$ | 2.874 | 0.033 | 86.72 | 2.870 | 0.014 | 200.8 |
| $\beta_8$ | 0.062 | 0.025 | 2.53 | 0.035 | 0.013 | 2.59 |

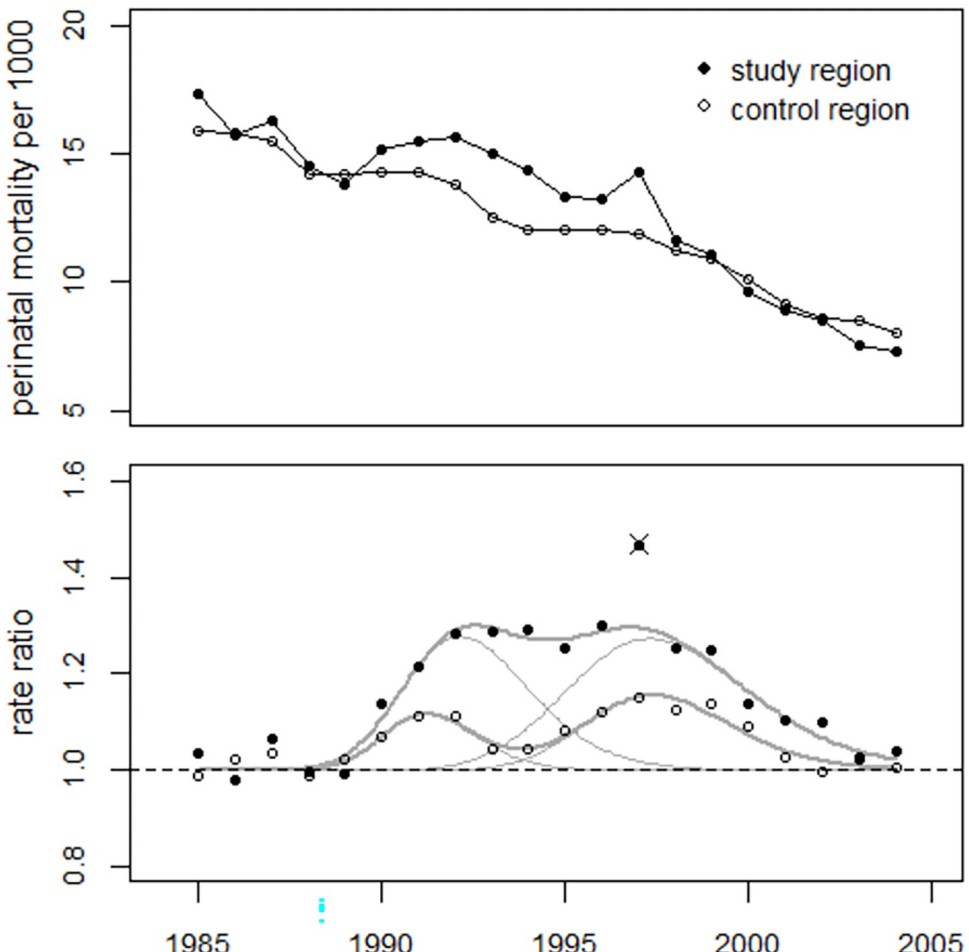

**Fig 5.** Upper panel: Development of perinatal mortality rates in the study and control regions. Lower panel: Ratio of observed to undisturbed predicted rates (rate ratios).

Table of S1 Dataset. To compare the development of perinatal mortality in the urban region of the city of Kyiv with the development in the more rural areas of the oblasts Zhytomyr and Kyiv, the results of a regression of the pooled data from the oblasts Zhytomyr and Kyiv were compared with the corresponding results for the city of Kyiv.

First, the pooled data from the oblasts Zhytomyr and Kyiv were analyzed. To obtain a more precise estimate of the trend parameter (slope) in the oblasts Zhytomyr and Kyiv, a combined regression of the data from the two oblasts with the data from the control region was carried out. Dummy variables were used to estimate possible effects in 1987. The regression yielded almost identical estimates for the trend parameters in the two regions ($-0.0369 \pm 0.0052$ vs. $-0.0368 \pm 0.0001$), so a common trend was used. The estimates for the variables characterizing the lognormal functions (medians and geometric standard deviations) were compatible with the margins of error. In 1987, a 10.9% increase in perinatal mortality was found in the two oblasts ($p = 0.055$) which compares to an increase of 3.5% ($p = 0.013$) in the control region. Fig 6 shows the trend in perinatal mortality rates in the two regions (upper panel) and the rate ratios (lower panel). The faint lines in Fig 6 indicate the two bell-shaped excess terms. When the observed perinatal deaths were used, the total number of excess perinatal deaths in the combined oblasts Zhytomyr and Kyiv was estimated at 1,164 (1,097 to 1,245).

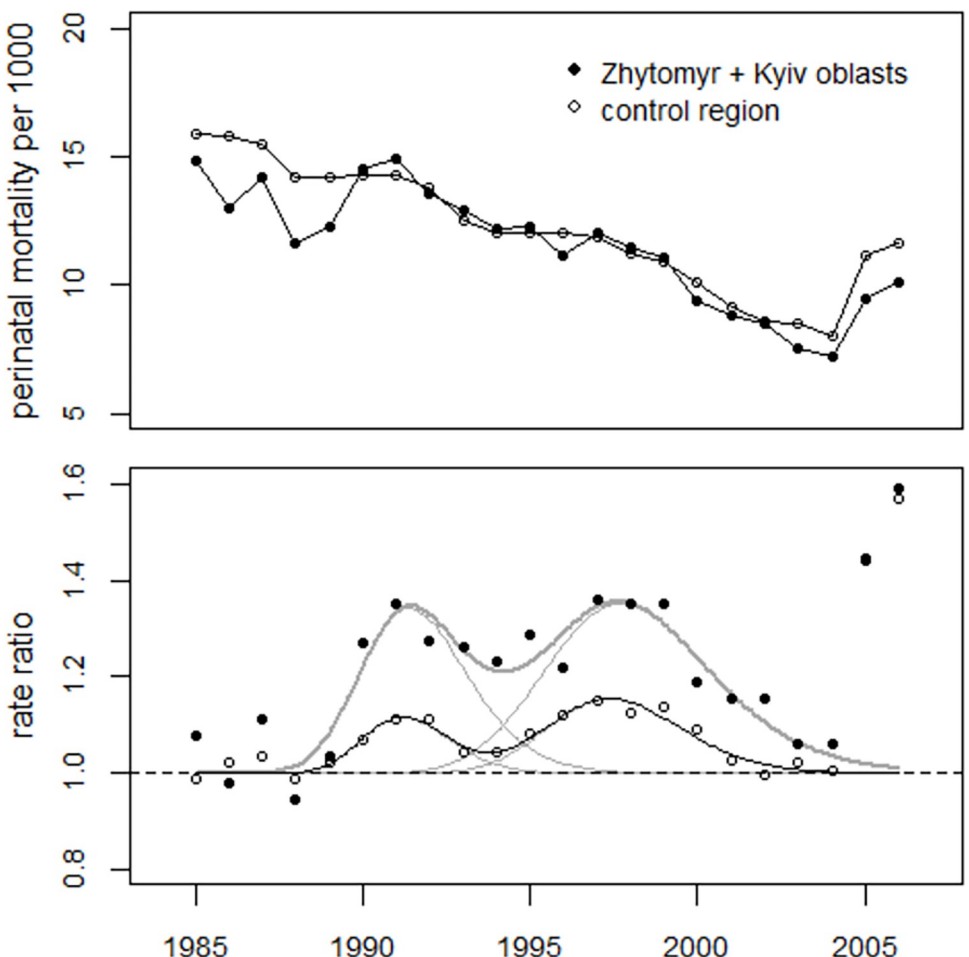

**Fig 6.** Upper panel: Development of perinatal mortality in the combined oblasts of Zhytomyr and Kyiv and in the control region. Lower panel: Relationship between observed and predicted undisturbed rates (rate ratios) and regression lines.

Perinatal mortality from the city of Kyiv was then analyzed. The deviance was 15.2 (df = 13), so the model fitted the data well. The trend parameter (slope) in the city of Kyiv was much steeper (-0.054 ± 0.003) than in the two rural oblasts (-0.037 ± 0.005), the difference was statistically significant (p = 0.006), see Table 6. The point in time of the first (1992.8) and

**Table 6. Regression results for perinatal mortality rates in the study and control regions.**

| Parameter | Kyiv City | | Zhytomyr + Kyiv oblast | | Control region | |
|---|---|---|---|---|---|---|
| | Estimate | SE | Estimate | SE | Estimate | SE |
| $\beta_0$ | -3.596 | 0.034 | -4.101 | 0.043 | -3.946 | 0.010 |
| $\beta_1$ | -0.0542 | 0.0029 | -0.0369 | 0.0052 | -0.0368 | 0.0009 |
| $\beta_3$ | 4.658 | 0.656 | 3.401 | 0.507 | 1.243 | 0.126 |
| $\beta_4$ | 2.560 | 0.015 | 2.455 | 0.024 | 2.431 | 0.014 |
| $\beta_5$ | 0.084 | 0.010 | 0.146 | 0.023 | 0.117 | 0.010 |
| $\beta_6$ | 6.907 | 0.897 | 5.429 | 1.355 | 2.527 | 0.263 |
| $\beta_7$ | 2.816 | 0.015 | 2.897 | 0.040 | 2.870 | 0.014 |
| $\beta_8$ | 0.032 | 0.049 | 0.103 | 0.050 | 0.035 | 0.013 |

second (1996.6) peak of the bell-shaped excess terms differ from those in the combined oblasts Zhytomyr and Kyiv (1991.4 and 1997.7) with p-values of p<0.001 and p = 0.066, respectively, for the differences. In 1990 and 1991, a sharp increase in perinatal mortality was detected in the combined oblasts Zhytomyr and Kyiv, while in the city of Kyiv, there was no deviation from the fitted model in these two years, see Fig 7. The number of excess perinatal deaths in the city of Kyiv for the period 1990–2004 was estimated at 759 (652 to 891), which corresponds to an overall increase of 19.5% (16.3 to 23.7%).

To determine any difference in effect size in 1990–2004 between the oblasts Zhytomyr and Kyiv, a combined regression of the oblast Zhytomyr, oblast Kyiv, and the control region—was carried out, A common time trend was assumed for the three regions, as the analysis of the pooled data from Kyiv and Zhytomyr showed no difference to the trend in the control region. Regression with individual trend parameters reduced the deviance not notably (p = 0.60) from 50.7 (df = 38) to 49.3 (df = 36). Dummy variables were used to estimate the increase in 1987. In Zhytomyr oblast, the estimates of the variables defining the bell-shaped excess terms, i.e. the position of the maxima and the width of the lognormal distributions, differed from those for the control region, while in Kyiv oblast they all matched within the error bounds. Regression with common parameters for these three variables yielded deviance = 52.2 (df = 41), p = 0.79.

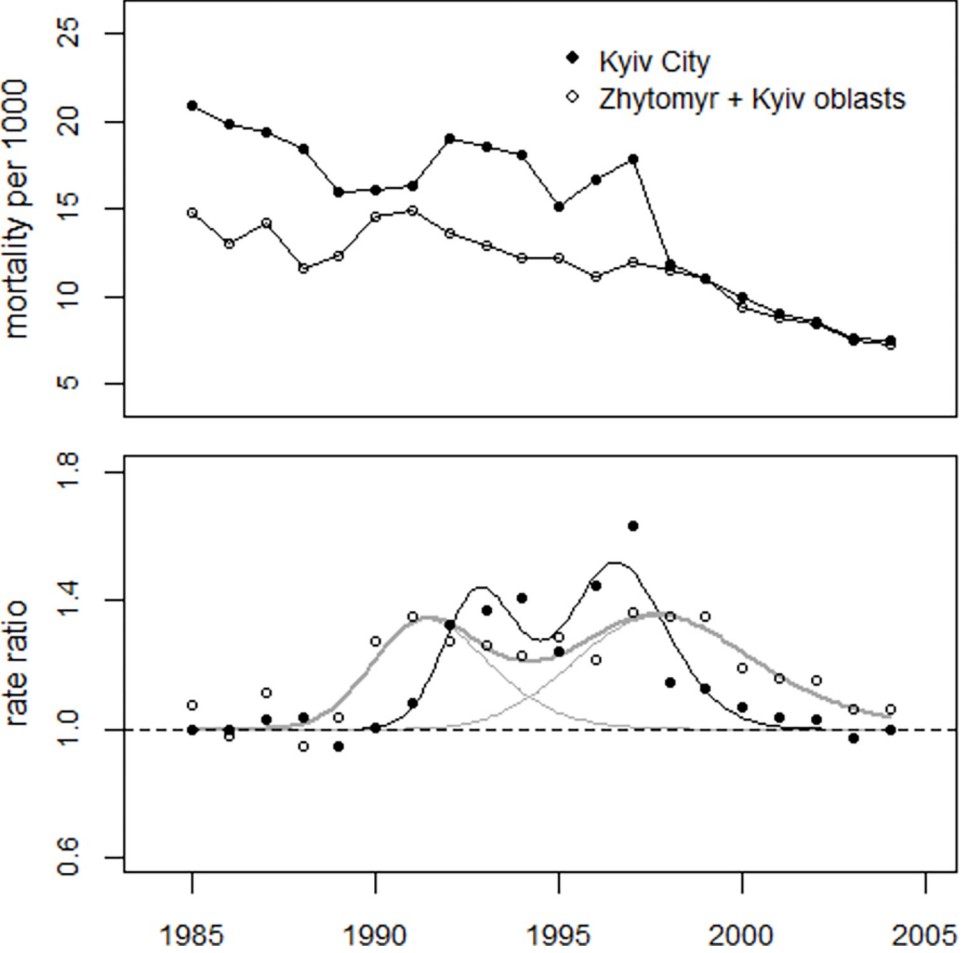

**Fig 7.** Upper panel: Development of perinatal mortality in the city of Kyiv and in the pooled oblasts Zhytomyr and Kyiv. Lower panel: Ratio between the observed and predicted undisturbed rates (rate ratios) and regression lines.

**Table 7. Results of combined regression of the data from the oblasts Zhytomyr and Kyiv.**

| Parameter | Zhytomyr oblast | | Kyiv oblast. | | Control region | |
|---|---|---|---|---|---|---|
| | Estimate | SE | Estimate | SE | Estimate | SE |
| $\beta_0$ | -4.213 | 0.096 | -4.014 | 0.050 | -3.946 | 0.010 |
| $\beta_1$ | -0.0338 | 0.0151 | -0.0414 | 0.0047 | -0.0368 | 0.0010 |
| $\beta_3$ | 4.903 | 1.218 | 2.404 | 0.648 | 1.243 | 0.128 |
| $\beta_4$ | 2.480 | 0.046 | 2.418 | 0.034 | 2.431 | 0.014 |
| $\beta_5$ | 0.185 | 0.046 | 0.107 | 0.030 | 0.117 | 0.011 |
| $\beta_6$ | 7.375 | 3.834 | 4.154 | 1.374 | 2.527 | 0.268 |
| $\beta_7$ | 2.937 | 0.106 | 2.873 | 0.038 | 2.870 | 0.014 |
| $\beta_8$ | 0.189 | 0.077 | 0.044 | 0.068 | 0.035 | 0.013 |

After removing five unnecessary variables, the final model improved the goodness of fit; it reduced the overdispersion from 1.37 to 1.27. The parameter estimates and standard errors (SE) resulting from regression with individual parameters for the three data sets are shown in Table 7.

The number of excess perinatal deaths in 1990–2004 was estimated at 964 in the Zhytomyr oblast, an overall increase of 49 percent. In the Kyiv oblast, 302 excess perinatal deaths were determined, an overall increase of 12 percent. In 1987, there was a significant increase of 21 percent in Zhytomyr oblast with 54 excess perinatal deaths (p = 0.013), while the increase in Kyiv oblast was much smaller (+6.4%, 22 excess deaths, p = 0.34). Fig 8 shows the trends of perinatal mortality rates in Zhytomyr and Kyiv oblasts and the respective regression lines.

The numbers of excess perinatal deaths in 1990–2004 by region are shown in Table 8.

## Analysis of the odds ratios

Regression of the ratios of perinatal mortality rates in the study region to the rates in the control region was performed using model (2). The effect of the bell-shaped excess term was highly statistically significant; the deviance was 81.5 (df = 18) without and 19.6 (df = 15) with the excess term (p<0.0001, F-test). The estimated number of excess perinatal deaths in 1990–2004 was 1,111 (747 to 2,719). Regression without the 1997 data reduced the deviance to 12.9 (p = 0.017) and yielded 856 (640 to 1,393) excess cases in 1990–2004. The trend in odds ratios is shown in Fig 9, together with the result of regression without the 1997 data.

A regression was then carried out using the strontium model (3). For the period 1990–1999, annual numbers of mothers aged 15 through 44 were available from the city of St Petersburg, see S5 Table in S1 Dataset. Three superimposed lognormal density distributions fitted the data well (see Fig 10).

To obtain the strontium concentration Sr(x) in pregnant women in year x after the Chornobyl accident, where x is calendar year minus 1986, the proportion of mothers who were 13 or 14 years old in 1986 is determined by the maternal age distribution. To adjust for strontium elimination, the model allowed for a 2.6% decrease in strontium burden per year [6].

Regression of the odds ratios with the strontium model and age 13 at exposure (in 1986) yielded deviance = 18.5 (df = 17). This compares with deviance = 81.6 (df = 18) without the strontium term. Assuming age 14 at exposure increased the deviance to a value of 24.6. With an age of 12 years at exposure, the deviance was 28.8.

Since the age at exposure was determined from the data, an F-test with two degrees of freedom has to be used to determine the significance of the strontium effect. With deviance = 81.6 (df = 18) without- and deviance = 18.5 (df = 16) with the strontium term, an F-test with two

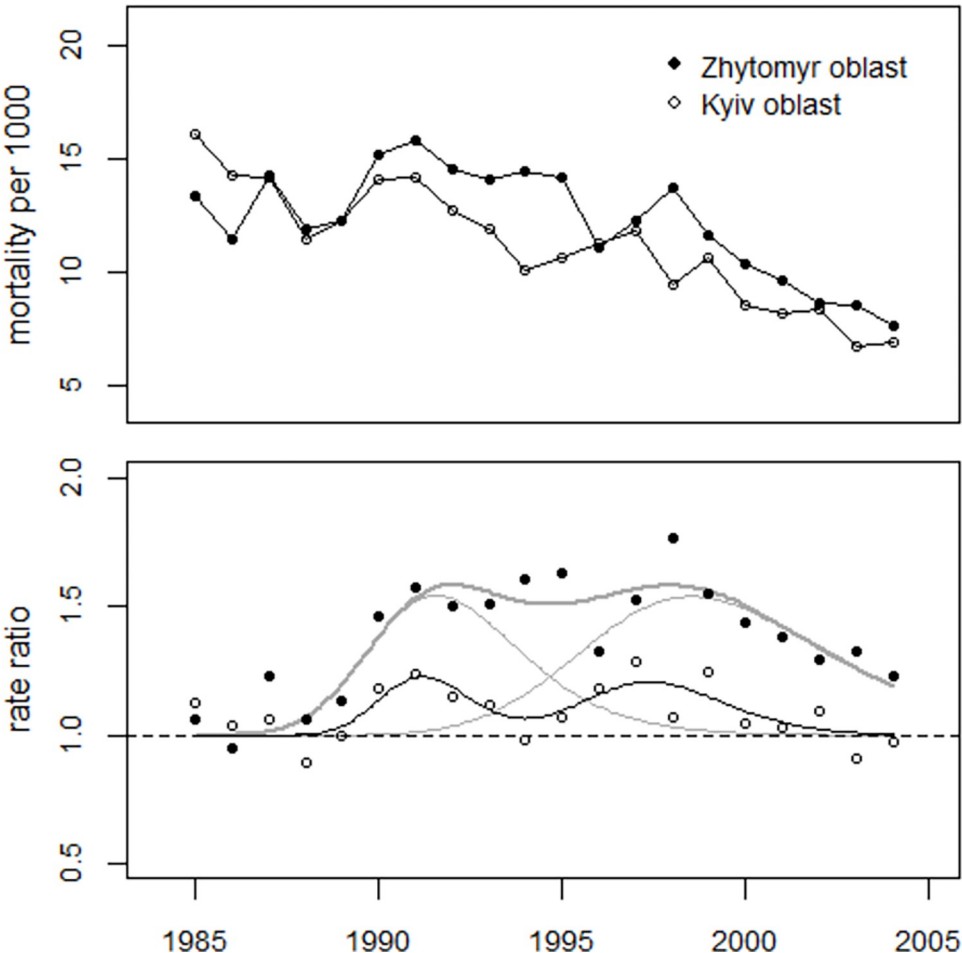

**Fig 8.** Upper panel: Trends in perinatal mortality rates in Kyiv and Zhytomyr oblasts Lower panel: Ratio between observed and predicted undisturbed rates (rate ratios) with regression lines.

degrees of freedom yielded p = 7E-6 for the effect of the strontium term. The number of excess perinatal deaths in 1990–2004 was estimated at 1,537 (1,229 to 1,740).

Regression without the 1997 data reduced the deviance from 18.5 (df = 17) to 12.8 (df = 16), p = 0.016. Regressions for ages 12 to 15 at exposure yielded deviances of 25.6, 12.8, 15.9, and 29.9 for ages 12, 13, 14, and 15, respectively.

Fig 11 shows the deviances of regressions at ages 12 to 15 at exposure in 1986. The dashed line shows the critical value of the deviance, i.e., the deviance for age 13 at exposure plus the

**Table 8. Observed (O) and predicted (E) perinatal deaths in 1990–2004 per region.**

| Oblast/region | O | E | O-E | O/E |
|---|---|---|---|---|
| Ukraine | 84,573 | 77,289 | 7,284 | 1.094 |
| Study region | 10,376 | 8,583 | 1,792 | 1.209 |
| Control region | 74,197 | 68,713 | 5,484 | 1.080 |
| Zhytomyr + Kyiv oblasts | 5,721 | 4,557 | 1,164 | 1.255 |
| Zhytomyr oblast | 2,914 | 1,950 | 964 | 1.495 |
| Kyiv oblast | 2,807 | 2,505 | 302 | 1.121 |
| Kyiv city | 4,655 | 3,906 | 749 | 1.192 |

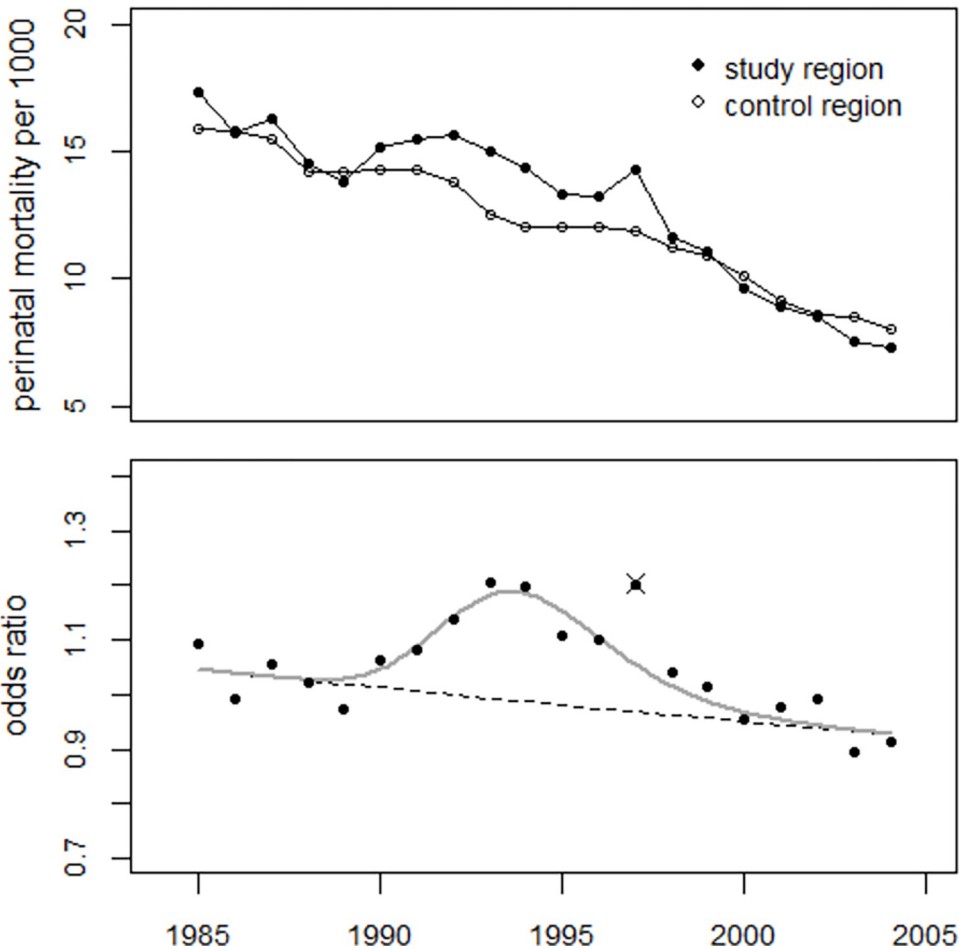

**Fig 9.** Upper panel: Trend in perinatal mortality rates in the study and control regions. Lower panel: Development of odds ratios and result of the regression with a bell-shaped excess term.

critical F value (here: 4.45). The intersections of the dashed line and the solid line mark the upper and lower bounds of the 95% confidence interval for age at exposure in 1986. The best estimate for age at exposure is 13.4 years, with a 95% confidence interval of 12.5 to 14.2 years.

Fig 12 shows the trend of the odds ratios in the study region and the result of regression with age 13 at exposure and without the 1997 data. The number of excess perinatal deaths in the period 1990–2004 was estimated to be 1,450 (1,183 to 1,706), corresponding to a mean odds ratio of 1.162 (1.129 to 1.197). The annual data of the odds ratios and the fitted values resulting from regressions with models (2) and (3) are shown in S4 Table of S2 File.

**Kyiv City.** Regressions of the odds ratios in Kyiv city with the model (2), i.e. with one bell-shaped excess term with and without the 1997 data, gave deviances of 32.8 (df = 15) and 17.4 (df = 14) respectively (p = 0.0035). Therefore, the 1997 data was considered an outlier and removed from the regression. The trend parameter (slope) of odds ratios, reflecting the difference in perinatal mortality trends between Kyiv city and the control region, was -0.018 (-0.024 to -0.012). The number of excess perinatal deaths in 1990–2004 was estimated at 484 (357 to 625) which is a mean increase of 11.6 (8.3 to 15.5) percent. The effect of the strontium term was highly statistically significant; the deviance was 91.0 (df = 17) and 26.9 (df = 16) resulting from the regressions without and with the strontium term (p<0.0001). The number of excess

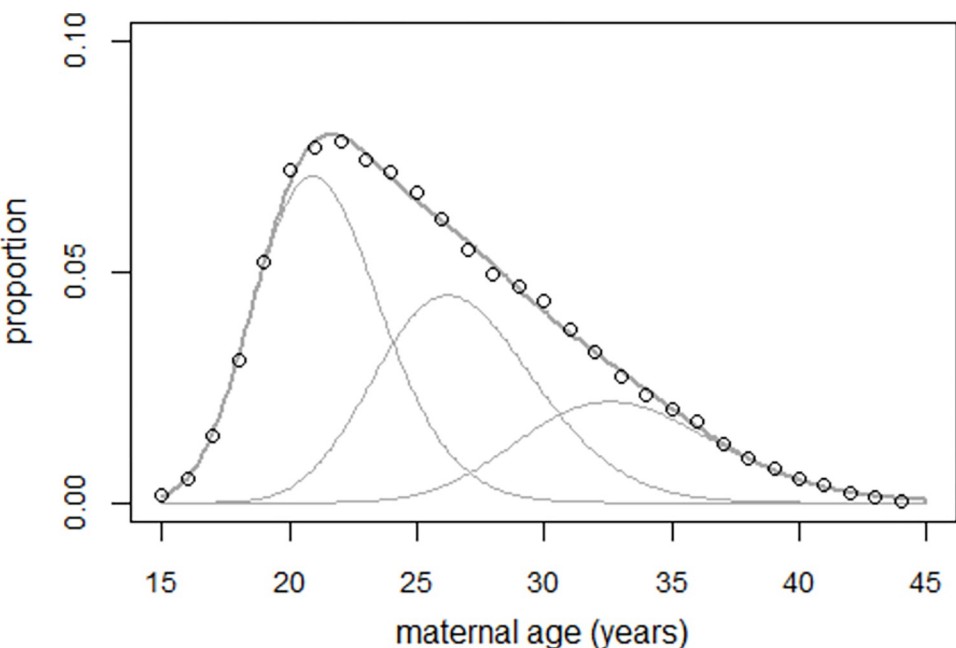

**Fig 10. Average age distribution of mothers in St. Petersburg in the period 1990–1999 and result of the regression with three superimposed lognormal density distributions.**

perinatal deaths during 1990–2004 was estimated at 946 (710 to 1,165) which is a mean increase of 25.5 (18.0 to 33.4) percent. The regression results are shown in Fig 13.

**Oblasts Zhytomyr and Kyiv.** Regressions of the odds ratios for the oblasts of Zhytomyr and Kyiv can be carried out for the period (1981–2006), as possible underreporting before

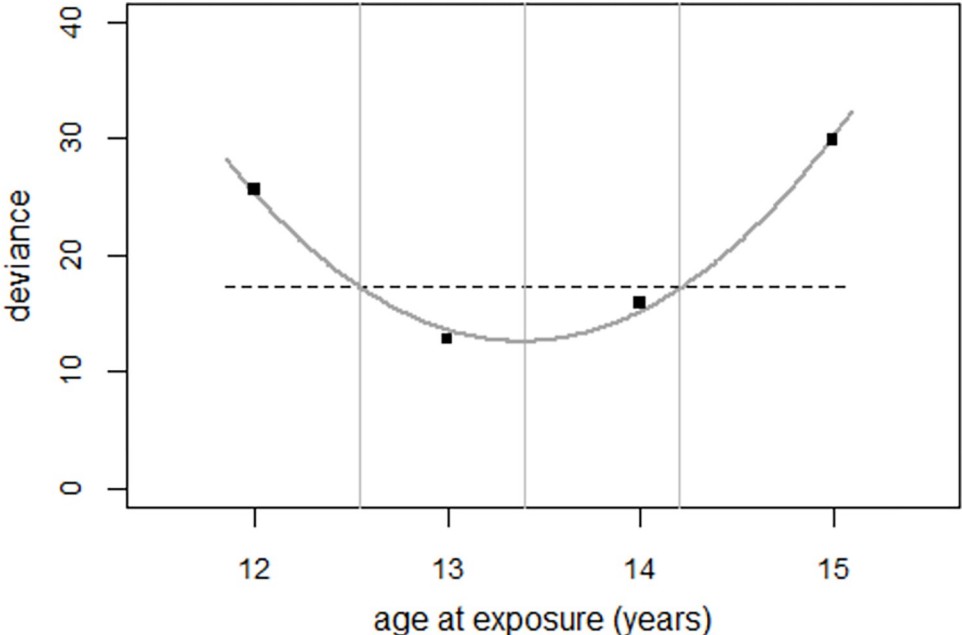

**Fig 11. Deviance as a function of the age at exposure in 1986 and regression line.** The horizontal dashed line shows the critical value of the deviance.

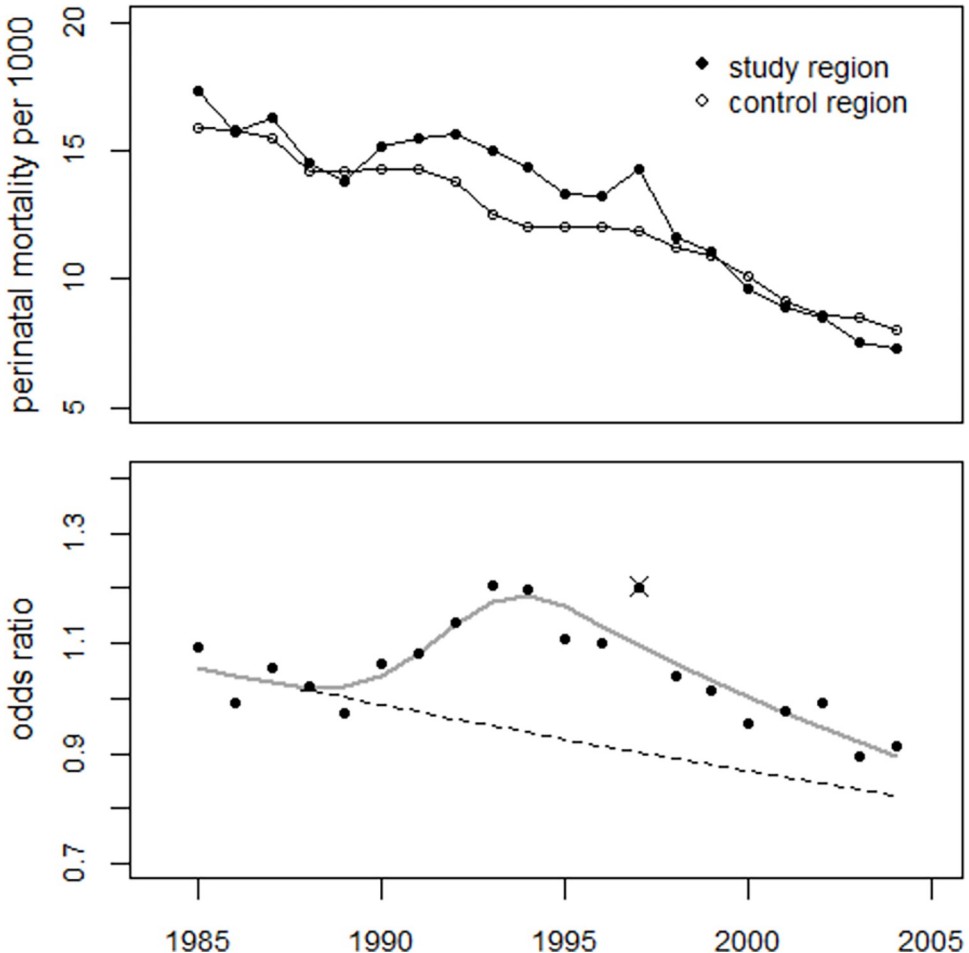

**Fig 12.** Upper panel: Trends in perinatal mortality rates in the study region and the control region. Lower panel: Observed odds ratios and result of the regression with the strontium model without the 1997 data. The broken line shows the predicted undisturbed trend of the odds ratios.

1985 and definitional changes after 2004 are likely to affect the data from both the study and control regions and are therefore likely to cancel each other out. To start with, a regression of the odds ratios of the pooled data from the oblasts of Zhytomyr and Kyiv was carried out. Regression model (2) was used but without a time variable, as the trend analysis had not revealed any difference in the slopes. A dummy variable for 1990–1991 was added to estimate the peak in these two years which was not observed in the data from the city of Kyiv. The model fitted the data well (deviance = 20.4, df = 21); the deviance without the excess terms was 65.4 (df = 25), so the effect of the excess terms was highly statistically significant (p<0.0001). The peak in the years 1990–1991 was also statistically significant (p = 0.009). The number of excess perinatal deaths in the years 1990–2004 was estimated at 643 (515 to 768) which corresponds to an average increase of 13 (10 to 16) percent.

The regression with the model (3), i.e., with the strontium term and age 13 at exposure in 1986, yielded a deviance of 20.9 (df = 23), comparable to deviance = 20.4 (df = 21) obtained with the model (2). The effect of the strontium term was highly statistically significant (p = 5E-6). The increase in 1990–1991 was also highly significant (p = 0.001). The number of excess perinatal deaths was estimated to be 643 (537 to 747), which corresponds to an overall increase

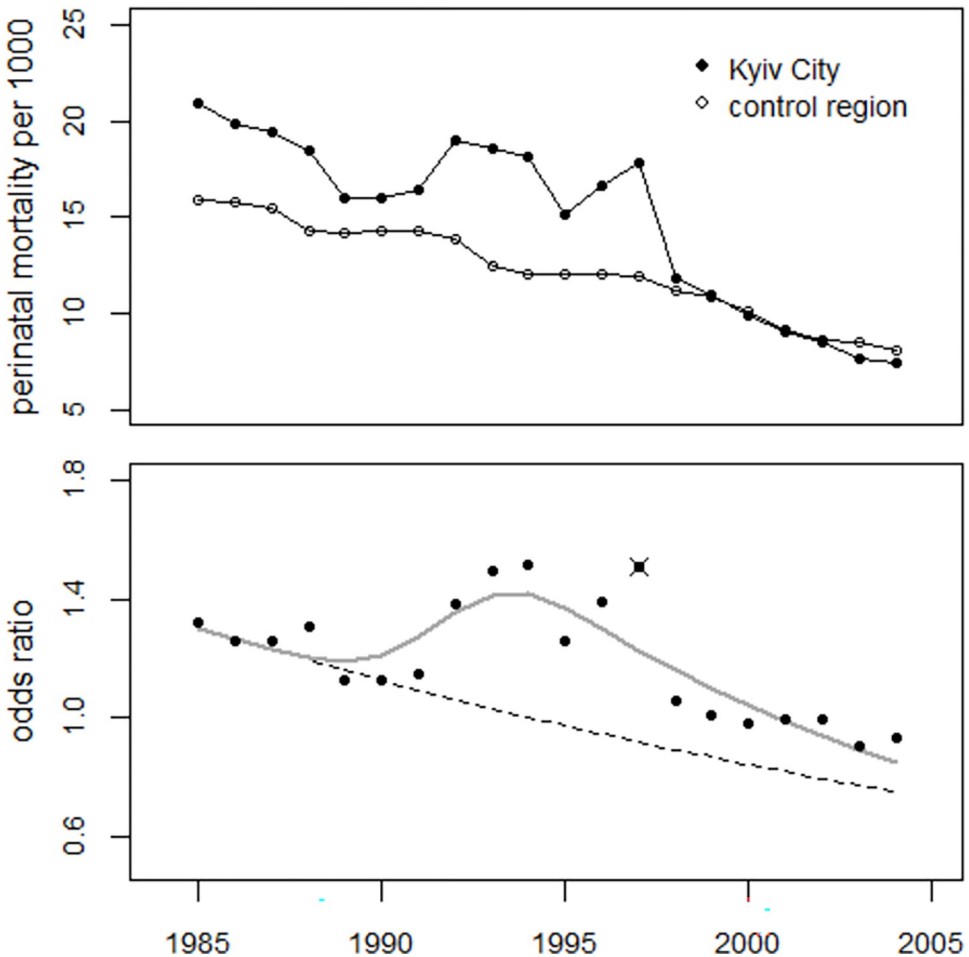

**Fig 13.** Panel A: Panel A: Trend of perinatal mortality rates in the city of Kyiv and in the control region. Panel B: Trend of odds ratios and result of the regression with the strontium model.

of 12.7 (10.4 to 15.0) percent. The trend of the odds ratios and the result of the regression with the strontium model are shown in Fig 14.

A possible explanation of the observed peak in 1990–1991 could be the difference in maternal age distribution between rural and urban regions. In the present study, the age distribution of mothers from St. Petersburg was used. However, this may not be a good proxy for the age distribution of mothers in more rural areas such as Zhytomyr and Kyiv oblasts where pregnancies tend to occur at a younger age than in cities.

A combined regression of the odds ratios of perinatal from the oblasts Zhytomyr and Kyiv was then performed. Common parameters were used for the medians and the geometric standard deviations of the bell-shaped excess terms. Dummy variables were used to estimate the increases in the years 1990–1991 and the deviations of the odds ratios in 2005 from the trend in both oblasts. The effect of the bell-shaped excess term was highly statistically significant in Zhytomyr oblast (p = 0.0005) and not notable in Kyiv oblast (p = 0.95), The difference in intercepts was negligible (p = 0.62), The regression results (parameter estimates, with standard errors (SE), t-values and p-values are shown in Table 9. In both oblasts, the peaks in 1990–1991 were not statistically significant, with increases of 11.5% in Zhytomyr (p = 0.17) and 9.6%

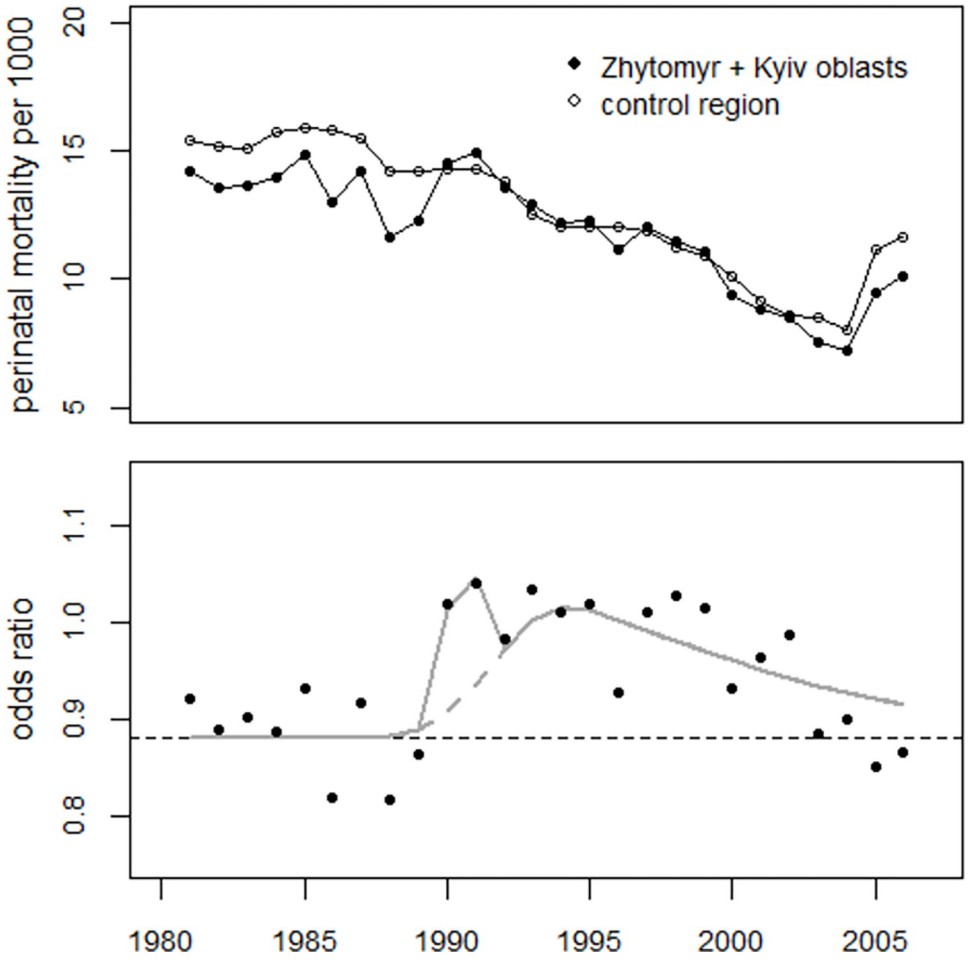

**Fig 14. Trend of perinatal mortality rates in the pooled oblasts Zhytomyr and Kyiv and the control region.** Panel B: Trend of odds ratios and result of the regression with the strontium model.

in Kyiv (p = 0.11). In 2005, there was a 17% increase in the Zhytomyr oblast (p = 0.030) and a 19% decrease in the Kyiv oblast (p = 0.003).

Regression with a common intercept, and without the bell-shaped excess term in Kyiv oblast, increased the deviance from 66.9 (df = 42) to 67.3 (df = 44), p = 0.87. The trend of perinatal mortality rates in the two oblasts and the trend of the odds ratios is shown in Fig 15. The number of excess perinatal deaths in 1990–2004 was estimated at 454 (381 to 525) in Zhytomyr oblast and at 48 (-33 to 128) in Kyiv oblast.

**Table 9. Regression results for odds ratios in the Zhytomyr and Kyiv oblasts.**

| Parameter | Zhytomyr oblast | | | | Kyiv oblast | | | |
|---|---|---|---|---|---|---|---|---|
| | Estimate | SE | t-value | p-value | Estimate | SE | | p-value |
| $\beta_1$ | -0.102 | 0.027 | -3.815 | 0.0004 | -0.059 | 0.011 | -5.298 | <0.0001 |
| $\beta_3$ | 3.391 | 0.898 | 3.774 | 0.0005 | 0.049 | 0.811 | 0.061 | 0.9516 |
| $\beta_4$ | 2.836 | 0.095 | 29.98 | <0.0001 | 2.836 | | | |
| $\beta_5$ | 0.265 | 0.106 | 2.493 | 0.0167 | 0.265 | | | |
| $\beta_6$ | 0.109 | 0.079 | 1.382 | 0.1742 | 0.092 | 0.057 | 1.616 | 0.1136 |
| $\beta_7$ | 0.154 | 0.069 | 2.241 | 0.0304 | -0.213 | 0.068 | -3.134 | 0.0031 |

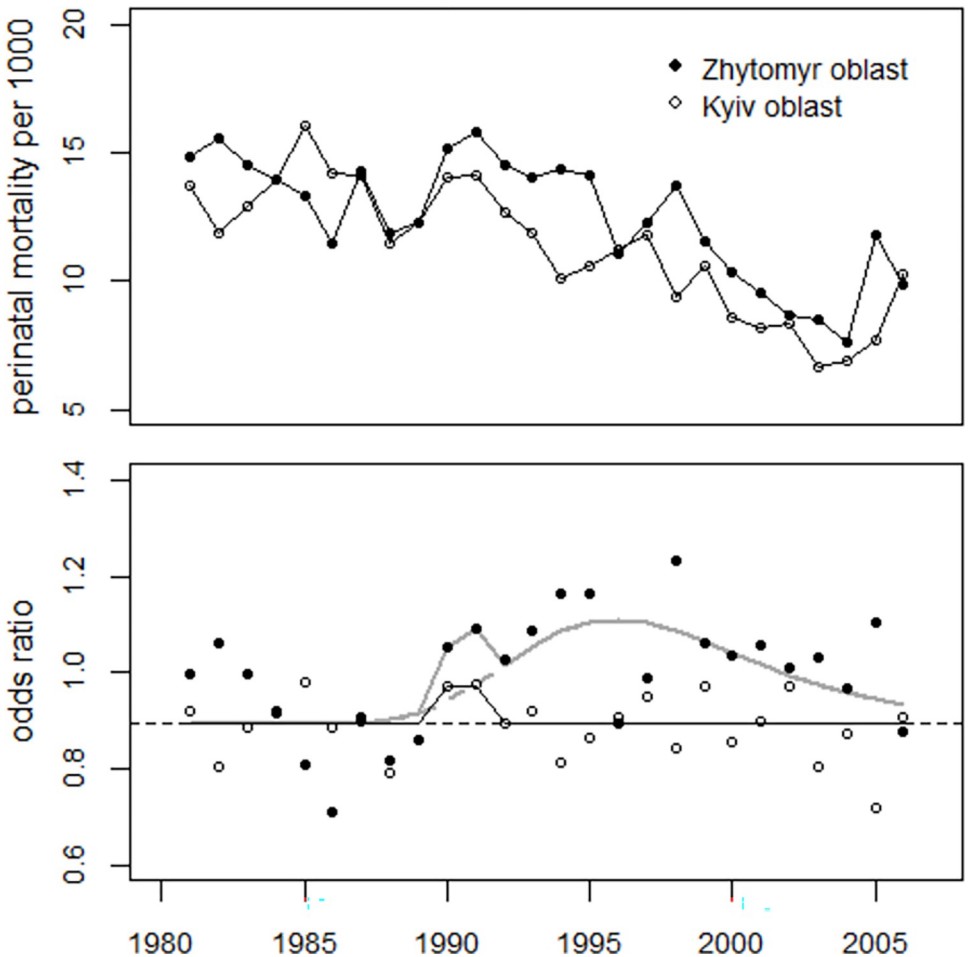

**Fig 15.** Upper panel: Perinatal mortality rates in the oblasts Zhytomyr and Kyiv. Lower panel: Trend of the odds ratios and regression lines. The faint horizontal line with the rise in 1990–1991 shows the regression result for Kyiv oblast.

Table 10 shows the number of observed (O) and predicted (E) perinatal deaths, the number of excess deaths (O-E), and their ratio (O/E) for the city of Kyiv, Kyiv oblast, Zhytomyr oblast, and the entire study region.

To determine the total number of Chornobyl-related perinatal deaths in Ukraine, the following approach is proposed. Let us assume that the first peak in the development of perinatal mortality can be attributed to radiation, while the second peak may be a consequence of the socio-economic crisis. In the trend analysis of the combined data from the oblasts Zhytomyr and Kyiv, the two peaks are well separated, see Fig 6. The number of excesses associated with

**Table 10. Excess perinatal deaths in 1990–2004 from odds ratio analysis.**

| Oblast/region | O | E | O-E | O/E |
|---|---|---|---|---|
| study region | 10,376 | 9,438 | 938 | 1.100 |
| Kyiv City | 4,655 | 4171 | 484 | 1.116 |
| Kyiv oblast | 2,807 | 2,759 | 48 | 1.018 |
| Zhytomyr oblast | 2,914 | 2,460 | 454 | 1.184 |
| Kyiv+Zhytomyr obl. | 5,721 | 5,102 | 619 | 1.121 |

the first and second excess terms are 632 and 564, respectively, based on the fitted values. The 632 excesses associated with the first peak are compatible with the 619 excesses determined in the odds ratio analysis of the odds ratios of this data, see Table 10.

For Ukraine as a whole, the trend analysis yielded 7,478 excess deaths in the period 1990–2004, of which 3,482 deaths were attributed to the first excess term and 3,996 to the second excess term. Thus, about 3,500 perinatal deaths may be attributed to the Chornobyl accident in Ukraine.

**Oblast Zhytomyr versus oblast Poltava.**   The oblast Poltava was classified as "clean", which means that the average Cs-137 surface contamination in 1986 was less than 37 kBq per square meter. Poltava oblast was often used as a control region in studies on perinatal mortality in Ukraine. In addition to the data from Ukraine and the oblasts of the study region, Professor Omelyanets also provided the author with perinatal mortality data from the Poltava oblast. They are presented in S2 Table of S1 Dataset. The odds ratios of perinatal mortality from The oblast Zhytomyr were analyzed with Poltava oblast as the control region. The regression with the model (3), supplemented with dummy variables for 1986 and 1991, showed an increasing time trend (+1.2% per year, p = 0.048), a significant effect of the strontium term (p = 0.022), a significant increase in 1991 (+23%, p = 0.021) and a significant decrease in 1986 (-23%, p = 0.007). The number of excess perinatal deaths in the period 1990–2004 was estimated at 412 (202 to 601), which compares with the 454 excess cases for Zhytomyr oblast in Table 10. The development of the odds ratios is shown in Fig 16.

**Oblast Rivne.**   During the review process, the question arose as to why the study region did not include the oblast Rivne, which was also heavily contaminated by the Chornobyl fall-out. However, the contamination was essentially due to cesium, whereas the present study examines the possible effects of strontium. The author had no information on strontium deposition in Rivne oblast, but since strontium is less volatile than cesium, it contaminated a much smaller around the Chornobyl NPP than cesium. A map of strontium deposition near the Chornobyl site can be found in Figure IV.1 of the Atlas of Caesium-137 contamination of Europe after the Chornobyl Accident [9], see S5 Fig of S1 File.

The data on perinatal mortality in the Rivne region were kindly provided to the author by OMNI-net Ukraine in Rivne (http://ukraineomni.org/en/about_eng/). They are shown in S3 Table of S1 Dataset. The trends in perinatal mortality in Rivne and the modified control region, i.e. Ukraine excluding the study region and Rivne oblast, are very similar. A linear regression of the ratio of the perinatal mortality rate in Rivne oblast to the rates in the control region (odds ratios) with only the strontium term for age 13 at exposure (Sr13) as the independent variable showed no notable effect of the strontium term (p = 0.63). The trend in perinatal mortality rates in the Rive oblast and the control region as well as the development of the odds ratios are shown in Fig 17.

**Comparison with data from Belarus.**   Finally, the result for the odds ratios in the Zhytomyr oblast will be compared with the corresponding results for the Gomel oblast in Belarus, which were published in [3]. The data from Belarus are shown in S4 Table of S1 Dataset. The study period was limited by the availability of data from Belarus (1985–1997). Belarus, excluding the Gomel oblast and the city of Minsk, was used as a control region for the Gomel and Zhytomyr oblasts. In Belarus, the definition of stillbirths was changed in 1994, which was not the case in Ukraine. Therefore, to adjust for the change in definition, the annual number of stillbirths in Zhytomyr oblast after 1993 was multiplied by the observed level shift in the trend of the stillbirth rate in Belarus from 1994 onwards. The strontium term used to analyze the Ukrainian data was also used for the analysis of the data from Belarus. A combined regression of the odds ratios in the two oblasts with a common intercept but individual parameters for the strontium term was performed. Dummy coding was used to identify possible effects in

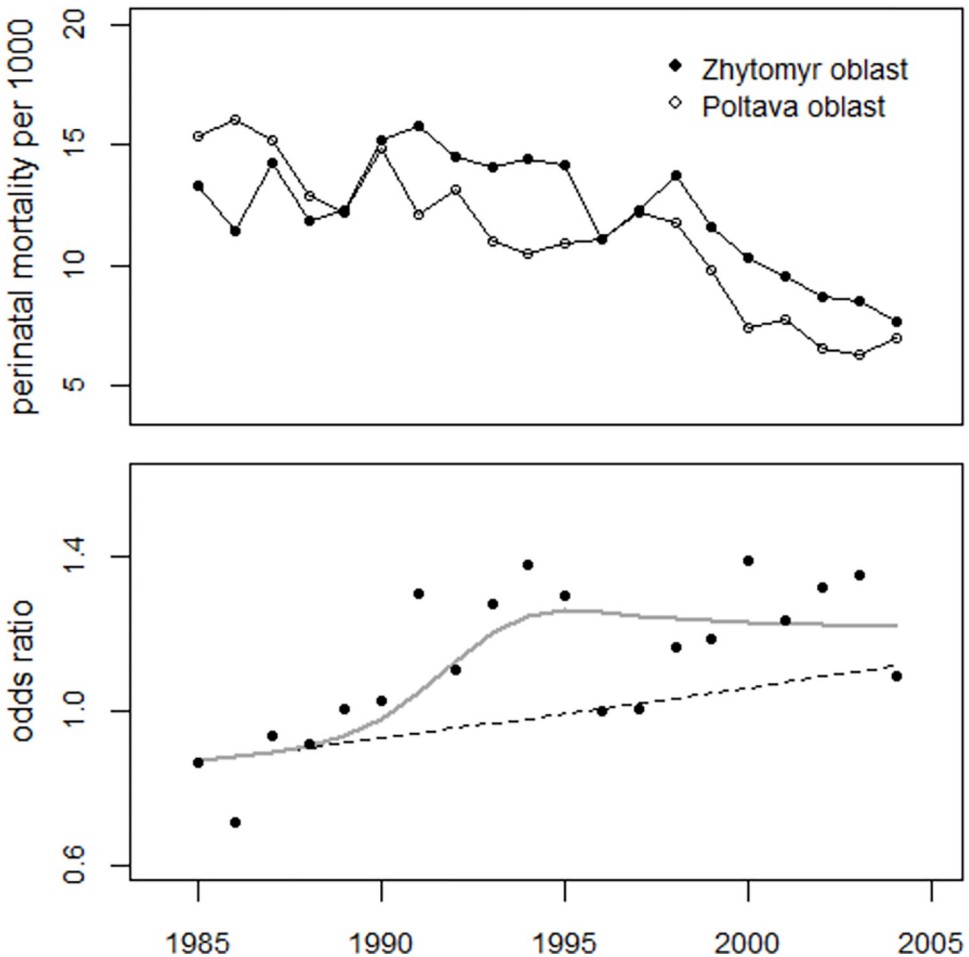

**Fig 16.** Upper panel: Development of perinatal mortality rates in the oblasts of Zhytomyr and Poltava. Lower panel: Odds ratios and result of regression with the strontium model (3).

1987 and in 1990–1991, where an additional peak was detected in the data of the Zhytomyr and Kyiv oblasts.

Regression with age 13 at exposure gave deviance 10.2 (df = 20) while regression with age 14 at exposure yielded deviance 7.8 (df = 20). Thus, both strontium terms fitted the data well; regression with only an intercept yielded a deviation of 116 (df = 25) so the improvement in fit with the full model was highly statistically significant. With age 14 at exposure, the strontium effect was 1.8 times greater in Zhytomyr than in Gomel. In 1987, the odds ratios were significantly increased in both oblasts at 22% in Zhytomyr and 17% in Gomel. The increases in the years 1990–1991 were also significant with exceedances of 13% in Zhytomyr and 11% in Gomel. A regression without the variables for 1990–1991 gave deviance = 17.8 (df = 22) so the increase in 1990–1991 is highly significant (p<0.001, F-test or p = 0.007 when a chisquare test was applied). Fig 18 shows the odds ratios in the oblasts of Zhytomyr and Gomel and the regression result for the strontium term.

## Discussion

This paper examines the development of the perinatal mortality rate in Ukraine before and after the Chornobyl accident in 1986. An unexpected increase was observed in the 1990s,

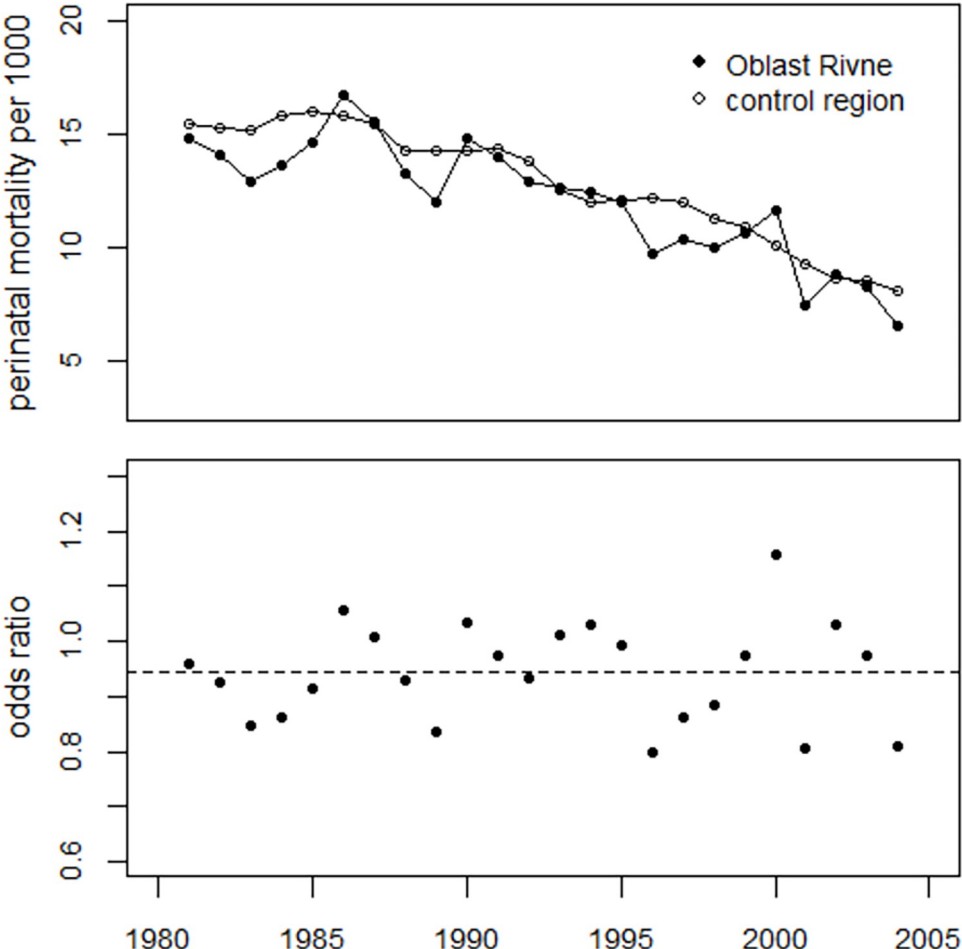

**Fig 17.** Upper panel: Development of perinatal mortality rates in the Rive oblast and in the control region. Lower panel: And development of odds ratios. The dashed line shows the mean value of the odds ratios.

which can be approximated by two bell-shaped terms superimposed on a long-term exponential trend. To examine a possible connection with the radiation exposure from the Chornobyl fallout, the development in the most contaminated oblasts of Ukraine (study region) was compared with the development in the rest of Ukraine (control region). The increase in the 1990s was almost three times as high in the study region as in the control region. The difference in the effect size between the study region and the control region was determined from the ratio of the perinatal mortality rates in the study region to the rates in the control region (odds ratio). The trend of the odds ratios shows an increase after 1989 with a maximum around 1994.

Regressions at an oblast level showed the highest strontium effect on perinatal mortality in Zhytomyr oblast in the period 1990–94, with an average increase of 52% and 672 additional perinatal deaths. In comparison, Ukraine as a whole recorded an increase of 9.4% and 3500 additional cases during this period. The peak in 1987 was statistically significant both in Zhytomyr oblast (+21%, 54 excess cases, $p = 0.013$) and in Ukraine as a whole (+3.8%, 435 excess cases, $p = 0.006$). While in this study the increase in the years 1990–1994 is attributed to strontium exposure, it is assumed that the increase in 1987 was mainly caused by the isotopes of cesium. Both effects are around five times higher in the oblast Zhytomyr than in Ukraine as a whole.

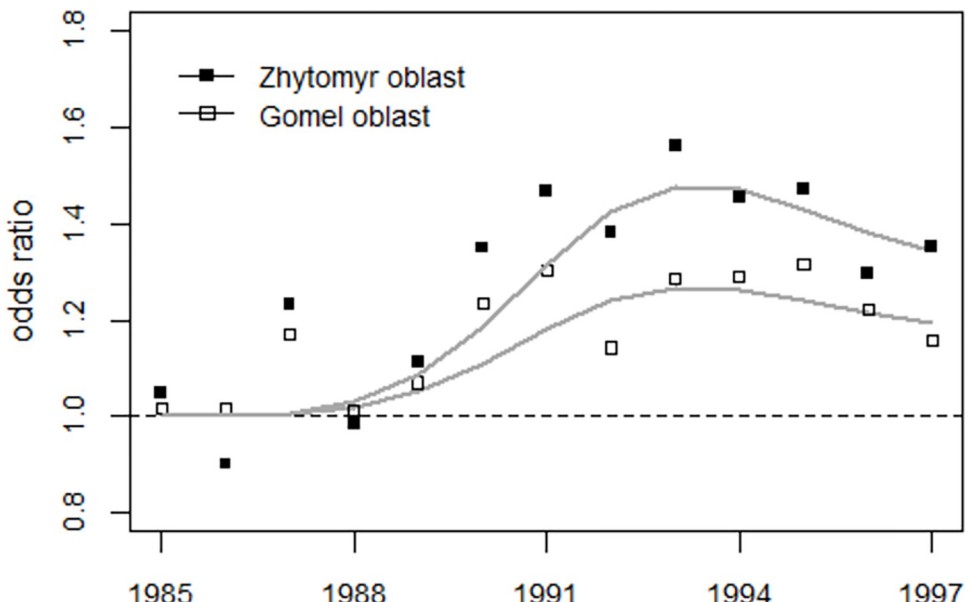

**Fig 18. Comparison of the development of the oddds ratios of perinatal mortality in the oblasts Zhytomyr (Ukraine) and Gomel (Belarus) and regression lines.**

The argument that the increase in the 1990s could be a consequence of the socioeconomic crisis following the collapse of the Soviet Union in 1991, rather than Chornobyl radiation, seems plausible. However, the socioeconomic crisis cannot explain the large difference in effect size between the study and control regions. Moreover, the increase in odds ratios started in 1990, before the effects of the socioeconomic crisis could be expected. Fig 19 shows the trends in perinatal mortality in Ukraine together with gross domestic product (GDP) per

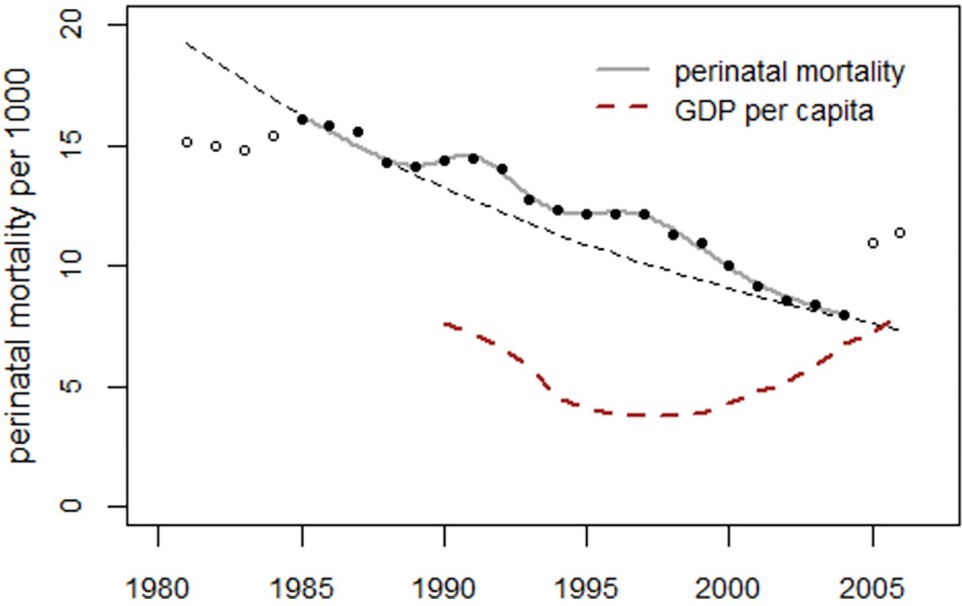

**Fig 19. Perinatal mortality rates in Ukraine and the development of gross domestic product (GDP) in units of USD 1000.**

capita (World Bank data [10]). GDP declined after 1991, reached a low around 1997, and rose steadily thereafter. While the second peak in the perinatal mortality trend appears to coincide with the trough in GDP per capita, there is no correlation between the first peak and the GDP trend.

An increase in perinatal mortality four years after the Chornobyl accident was not expected given the observed decrease in ambient radiation and food contamination in the post-1986 period. However, the increase is consistent with the observed increase in perinatal mortality rates in the 1990s in the Gomel region of Belarus [3]. The increase in Gomel oblast was interpreted in [3] as a late effect of strontium-90. According to [6], strontium incorporation is greatest during the period of increased bone growth, which occurs in girls aged 13 or 14 years. Strontium impairs the immune system, which in turn may be responsible for the increased perinatal mortality for mothers who ingested strontium from Chornobyl fallout as teenagers in 1986. In the present study, the best estimate of age at exposure (i.e. in 1986) was 13.3 (12.5 to 14.2) years.

There is no information on strontium concentration in pregnant women, but it can be assumed that it was higher in the most contaminated areas of Ukraine than in the rest of the country. Strontium is absorbed mainly through the consumption of bread, dairy products, and meat [11]. After Chornobyl, in the fall of 1986, highly contaminated meat was mixed in a ratio of 1:10 with normal meat to produce canned meat and sausages, which were widely used throughout the Soviet Union [12,13]. There was no real control region in Ukraine, as the contaminated food was consumed everywhere.

According to the International Commission on Radiological Protection [14], teratogenic radiation effects, such as increased perinatal mortality, are not expected to occur below a threshold dose of 100 mSv. However, even in the most contaminated oblasts of Ukraine, the estimated radiation doses to the population in the first year after the Chornobyl accident were only a few mSv (see Table 2 in [1]).

The present study is ecological; individual exposure data were not available. However, to study small health effects, the study population must be large enough to achieve the required statistical power. Case-control or cohort studies are not economically feasible for large populations, so the only practical way to detect small health effects is to conduct ecological studies, but these are inherently subject to limitations.

## Conclusion

It is suggested that the observed increase in perinatal mortality in the 1990s may be a late effect of incorporated radioactive strontium-90 on the immune system of pregnant women. The analysis is based on a theoretical model because no data on strontium concentrations were available; therefore, the results should be interpreted with caution. Similar studies in Ukraine's neighboring countries, especially Belarus, are recommended to substantiate the results.

## Supporting information

**S1 File. Supporting material.**
(DOCX)

**S2 File. Regression results.**
(DOCX)

**S1 Dataset. Data files.**
(DOCX)

## Acknowledgments

The present study would not have been possible without the knowledge about the properties of strontium gathered by the researchers at the Ural Research Center for Radiation Medicine in Chelyabinsk (South Urals). The author thanks Professor Omelyanets, Radiation Medicine Research Center AMS of Ukraine, for providing the data on perinatal mortality in Ukraine used in this study. Thanks also to Natalia Kovaleva for providing data on the age distribution of mothers from Saint Petersburg.

## Author Contributions

**Conceptualization:** Alfred Körblein.

**Data curation:** Alfred Körblein.

**Formal analysis:** Alfred Körblein.

**Investigation:** Alfred Körblein.

**Methodology:** Alfred Körblein.

**Project administration:** Alfred Körblein.

**Resources:** Alfred Körblein.

**Software:** Alfred Körblein.

**Supervision:** Alfred Körblein.

**Validation:** Alfred Körblein.

**Visualization:** Alfred Körblein.

**Writing – original draft:** Alfred Körblein.

**Writing – review & editing:** Alfred Körblein.

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
