## [Decision Letter · Decision Letter 0]

9 Jan 2024

PONE-D-23-37340Perinatal mortality after Chornobyl in contaminated regions of UkrainePLOS ONE

Dear Dr. Körblein,

Thank you for submitting your manuscript to PLOS ONE. After careful consideration, we feel that it has merit but does not fully meet PLOS ONE’s publication criteria as it currently stands. Therefore, we invite you to submit a revised version of the manuscript that addresses the points raised during the review process. In other words, congratulations! Both reviewers felt your paper had merit but needed a little more work before final acceptance. I think perhaps the two most important issues concern the fact that Sr-90 is not the only radioisotope of interest given that Cs-137 likely also  had a significant contribution to the received dose. Please explore this question as best you can. And of course there were many other isotopes released some of which have been well documented to be important for public health (e.g. iodine isotopes). Including some discussion beyond Sr likely broadens the audience for your paper.The other issue that must be addressed concerns the position of the Rivne Oblast in the analyses. There have been a number of papers on related topics indicating that because of soil type and other factors, Rivne Oblast may not be a control area and in fact might be considered to be a region of significant dose. Perhaps you can explore this question in your analyses. I think this could be very helpful for the broader discussion. 

There are several other issues that are presented by reviewer #2 that should be addressed in your revision. For example, is quite common to include as co-authors people who have contributed unpublished data. I am not sure to what extent this should be considered here. It is not a simple issue but please reflect on this as you prepare your revision. And of course, should you have any questions, feel free to contact me directly. 

We look forward to receiving your revised manuscript.

Kind regards,

Tim A. Mousseau

Academic Editor

PLOS ONE

2. We note that you have referenced (ie. 7. Omelyanets N et al. [7]) which has currently not yet been accepted for publication. Please remove this from your References and amend this to state in the body of your manuscript: (ie “Omelyanets N et al. [Unpublished]”) as detailed online in our guide for authors

Reviewers' comments:

Reviewer's Responses to Questions

**Comments to the Author**

1. Is the manuscript technically sound, and do the data support the conclusions?

Reviewer #1: Yes

Reviewer #2: Partly

2. Has the statistical analysis been performed appropriately and rigorously? 

Reviewer #1: Yes

Reviewer #2: Yes

3. Have the authors made all data underlying the findings in their manuscript fully available?

Reviewer #1: Yes

Reviewer #2: Yes

4. Is the manuscript presented in an intelligible fashion and written in standard English?

Reviewer #1: Yes

Reviewer #2: No

5. Review Comments to the Author

Reviewer #1: The authors have excellently inferred based on theroretical modelling that the observed increase in perinatal mortality in the 1990s may be a late effect of incorporated radioactive strontium-90 on the immune system of pregnant women. Though the results are not fully conclusive and further studies may be warranted in future, the manuscript will lay the foundation for such research and is hence acceptable for publication as it is. The reviewer congratulates the author for this thought provoking study.

Reviewer #2: It is self-evident that the author is a competent investigator and has produced a thorough report. However, from a biologic point of view, statistical analyses cannot improve data flaws nor interpretation flaws. We believe that the report focuses attention on an important issue - death as an end point of radiation "sickness" "syndrome". Incorporated radionuclides impact physiology and homeostasis and may enhance lethal effects of extemporaneous causes rendering infants more label, ultimately reflected in mortality. Incorporation of radionuclides is gradual by women, pregnant women, and by their infants. Also, noteworthy is that at the time of the reported peak of mortality, in Ukraine, the definitions of stillbirth-livebirth were modified. In summary, we believe this report is important and that the author or authors can easily restructure the presentation. In this spirit we offer the following suggestions.

1. The author indicates that the analysis of the data provided by Dr. N. Omelyanets was to be published jointly. The author submits this report without Dr. N. Omelyanets and instead acknowledges his contribution. A comment clarifying this discrepancy would be welcome.

2. The control data includes Rivne oblast which in turn includes Rivne Polissia. This region is considered by multiple experts to be among the most polluted by Chornobyl radionuclides. Furthermore, the soils in Rivne Polissia have the highest transfer index of radionuclides from soil to biota (see reports by Likhtarev et al.). This fact enhances the biologic impacts of incorporated radionuclides in this area. Rivne Polissia should not be included as a component of a "control" population.

3. The incorporation of radionuclides represents admixtures. Generally, the proportion of incorporated Cs-137 is greater than of Sr-90. The reasons why the author is entirely focused on one radionuclide is at best questionable. Perhaps, there are good reasons for this emphasis and if so, they should be stated more clearly.

4. The report concerns mortality and, as is stands, it almost conveys the notion of a statistical exercise which belongs in a supplement.

6. PLOS authors have the option to publish the peer review history of their article (what does this mean?). If published, this will include your full peer review and any attached files.

Reviewer #1: No

Reviewer #2: No

---

## [Author Response · Author response to Decision Letter 0]

26 Mar 2024

In my revised manuscript I tried at address all queries of the editors and the reviewers.

---

## [Editor Report · Decision Letter 1]

25 Apr 2024

Perinatal mortality after Chornobyl in contaminated regions of Ukraine

PONE-D-23-37340R1

Dear Dr. Körblein,

We’re pleased to inform you that your manuscript has been judged scientifically suitable for publication and will be formally accepted for publication once it meets all outstanding technical requirements. Congratulations. This is a valuable contribution to this field and you are to be commended for your painstaking effort to unravel the possible causes of variation in the data. 

Kind regards,

Tim A. Mousseau

Academic Editor

PLOS ONE

---

## [Editor Report · Acceptance letter]

8 May 2024

PONE-D-23-37340R1 

PLOS ONE

Dear Dr. Körblein, 

I'm pleased to inform you that your manuscript has been deemed suitable for publication in PLOS ONE. Congratulations! Your manuscript is now being handed over to our production team.

Kind regards, 

on behalf of

Dr. Tim A. Mousseau 

Academic Editor

PLOS ONE